# The chromatin remodeller CHD4 regulates transcription factor binding to both prevent activation of silent enhancers and maintain active regulatory elements

Andria Koulle[1,2†], Oluwaseun Ogundele[1,2], Devina Shah[1,2‡], India Baker[1], Maya Lopez[1], David Lando[1,2], Nicola Reynolds[1,3], Ramy Ragheb[1,3], Ernest D Laue[1,2], Brian Hendrich[1,2,3*]

[1]Cambridge Stem Cell Institute, Jeffrey Cheah Biomedical Centre, University of Cambridge, Cambridge, United Kingdom; [2]Department of Biochemistry, University of Cambridge, Cambridge, United Kingdom; [3]Living Systems Institute, University of Exeter, Exeter, United Kingdom

**\*For correspondence:**
B.D.Hendrich@exeter.ac.uk

**Present address:** †Department of Life Sciences, Imperial College London, London, United Kingdom; ‡University of Copenhagen, Novo Nordisk Foundation Center for Protein Research, Copenhagen, Denmark

**Competing interest:** The authors declare that no competing interests exist.

## eLife Assessment

This work offers **important** insights into the protein CHD4's function in chromatin remodeling and gene regulation in embryonic stem cells, supported by extensive biochemical, genomic, and imaging data. The use of an inducible degron system allows precise functional analysis, and the datasets generated represent a key resource for the field. The revised study offers **compelling** evidence and makes a significant contribution to understanding CHD4's role in epigenetic regulation. This work will be of interest to the epigenetics and stem biology fields.

**Abstract** Chromatin organisation and transcriptional regulation are tightly coordinated processes that are essential for maintaining cellular identity and function. ATP-dependent chromatin remodelling proteins play critical roles in control of genome structure and in regulating transcription across eukaryotes. Their essential nature, however, has made it difficult to define exactly how these functions are mediated. The chromatin remodeller CHD4 has been shown to be capable of sliding nucleosomes in vitro, and to regulate chromatin accessibility and gene expression in vivo. Using an inducible depletion system, here we identify a second mechanism of action for CHD4 in actively restricting the residence time of transcription factors (TFs) on chromatin. Together, these activities result in distinct, context-dependent outcomes: at highly accessible regulatory elements, CHD4 limits TF binding to maintain regulatory function, while at low-accessibility euchromatic regions, it prevents TF engagement and sustains chromatin compaction, thereby silencing cryptic enhancers. Collectively, these mechanisms enable CHD4 to reduce transcriptional noise while preserving the responsiveness of active regulatory networks.

## Introduction

Cell state transitions are driven by the activity of transcription factors (TFs). Many TFs bind to specific DNA motifs within chromatin, but the frequency of these motifs within mammalian genomes far

outnumbers sites at which protein binding is detectable. The ability of a TF to recognise its cognate motif is influenced by how the DNA encoding that motif is packaged in chromatin. Accessible sites, that is those associated with a low density of intact nucleosomes, are more likely to be identified and bound by TFs than are inaccessible sites, that is those associated with higher nucleosome density. Controlling chromatin accessibility is therefore crucial for controlling TF binding to cognate sites in regulatory regions. TF-binding patterns will then determine which gene regulatory regions are used to drive gene expression and thereby define cell identity.

Vertebrate cells contain multiple proteins capable of using energy derived from ATP hydrolysis to remodel nucleosomes. These chromatin remodellers share a conserved sucrose non-fermentable 2 (SNF2) helicase-like ATPase domain but otherwise have various additional functional domains which impact how they organise chromatin, transcription and DNA repair (*Hota and Bruneau, 2016*; *Narlikar et al., 2013*). Chromatin remodellers play important roles in mammalian development, and heterozygous mutations in the genes encoding them underlie a variety of developmental disorders in humans (*Gourisankar et al., 2024*; *Hota and Bruneau, 2016*; *Pierson et al., 2019*). Similarly, somatic mutations in chromatin remodeller subunit genes are increasingly being implicated in cancer initiation and/or progression (*Lai and Wade, 2011*; *Martincorena et al., 2017*; *Wilson and Roberts, 2011*).

CHD4 is an abundant ATP-dependent chromatin remodelling protein which plays important roles in chromatin organisation and cell fate decisions in many different aspects of metazoan development. Depletion of CHD4 in mouse or *Drosophila* cells, or overexpression of a dominant negative form of the protein, leads to increased chromatin accessibility at regulatory elements and DNase hypersensitive sites (*de Dieuleveult et al., 2016*; *Morris et al., 2014*; *Moshkin et al., 2012*), indicating that it predominantly functions to compact chromatin. Its impact on gene expression is less clear-cut, however. One study found that while CHD4 acted to repress genes with bivalent promoters, its activity at active promoters (marked with H3K4Me3) mainly facilitated transcription in mouse embryonic stem (ES) cells (*de Dieuleveult et al., 2016*). Another study found CHD4 controlled the probability of gene expression, rather than levels, during the first cell fate transition in mammalian embryogenesis (*O'Shaughnessy-Kirwan et al., 2015*). Several studies in somatic lineages have shown that CHD4 prevents lineage inappropriate gene expression during cell fate decisions in both mice and *Drosophila* (*Arends et al., 2019*; *Aughey et al., 2023*; *Gómez-Del Arco et al., 2016*; *Sreenivasan et al., 2021*; *Wilczewski et al., 2018*; *Yoshida et al., 2019*). Exactly how CHD4 has this varied impact on chromatin and transcription has not been defined.

CHD4 is the predominant chromatin remodelling subunit of the Nucleosome Remodelling and Deacetylation (NuRD) complex (*Wade et al., 1998*; *Xue et al., 1998*; *Zhang et al., 1998*). NuRD is a highly abundant chromatin remodeller present at active enhancers and promoters in many cell types. Its activity is important not only to regulate transcription but also the movement of enhancers in 3D space (*Basu et al., 2023*), and to maintain genome integrity (*Polo et al., 2010*; *Smeenk et al., 2010*). NuRD has been shown to exert two functions in mammalian cells: one is to control the nucleosome density at active enhancers, thereby regulating TF binding, enhancer activity and transcriptional output (*Bornelöv et al., 2018*; *Pundhir et al., 2023*), while the other is to prevent low-level, inappropriate transcription across the genome (*Burgold et al., 2019*; *Montibus et al., 2024*; *Ragheb et al., 2020*; *Saotome et al., 2024*).

Although mouse ES cells can survive with a complete loss of the histone deacetylase subunit of NuRD (*Burgold et al., 2019*), loss of the chromatin remodelling component, CHD4, leads to cell death (*Stevens et al., 2017*). While we have used genetics to create NuRD-low or NuRD-null ES cells in the past to define NuRD function, in both cases, the remodelling subcomplex (CHD4, GATAD2A/B, and CDK2AP1) remained on chromatin and the extent to which it could continue to remodel chromatin is not known (*Bornelöv et al., 2018*). As with any genetic change, it is also very difficult to know to what extent constitutive loss of protein activity has resulted in selection of cells using some compensatory mechanism to remain viable. CHD4 is also known to function outside of the NuRD complex (*O'Shaughnessy-Kirwan et al., 2015*; *Ostapcuk et al., 2018*; *Williams et al., 2004*). Assessing the direct function of NuRD's remodelling activity is therefore difficult as knockout of the remodeller results in a non-viable cell, while knockdown or exogenous overexpression of a dominant negative version of the remodeller (*Bornelöv et al., 2018*; *Feng and Zhang, 2001*) will likely produce a heterogeneous mix of cells with varying levels of remodeller activity and displaying increasing degrees of cell cycle arrest and apoptosis over time.

The CHD4/NuRD function described at enhancers has been defined both in genetic mutants and over a time course of NuRD reintroduction to null cells (*Bornelöv et al., 2018*; *Reynolds et al., 2012*). In contrast, the noise reduction function has been described in genetically NuRD-deficient or NuRD-null cells or after NuRD component knockdown (*Burgold et al., 2019*; *Montibus et al., 2024*; *Ragheb et al., 2020*; *Saotome et al., 2024*), meaning that this could either be a primary function of NuRD or a downstream consequence and/or cell adaptation of NuRD deletion/depletion. For these reasons, we have employed a degron system, which allows us to acutely deplete CHD4 protein in mouse ES cells and assess the consequences to chromatin and gene expression within minutes to hours of protein depletion, long before the cells begin to exhibit cell cycle defects.

## Results

### CHD4 has an immediate and widespread impact on chromatin accessibility

NuRD in mouse ES cells can exist in various forms and it can contain different paralogs of key components such as MBD2 or MBD3, GATAD2A or GATAD2B, HDAC1 or HDAC2, and MTA1, 2 and/or MTA3 (*Reid et al., 2023*). In ES cells, NuRD has only one chromatin remodelling subunit: CHD4. We therefore created ES cell lines in which the endogenous *Chd4* alleles were homozygously tagged with a mini-Auxin-inducible degron (mAID) (*Nishimura et al., 2009*). CHD4-mAID was largely depleted from both the nucleoplasm and chromatin within 60 min of Auxin addition (*Figure 1A*). CHD4-depleted cells showed a normal cell cycle profile until 24 hr of depletion, when they began to undergo cell cycle arrest at the G1/S checkpoint (*Figure 1B*). By 30 hr CHD4-depleted cells were undergoing apoptosis, which increased through 48 hr. We therefore focused our analyses on the first few hours after CHD4 depletion, well before cells started to undergo cell cycle arrest.

CHD4 depletion had an immediate and widespread impact on chromatin accessibility, as measured by calibrated ATAC-seq. After 60 min of CHD4 depletion, there were more than 13,000 sites showing a significant change in accessibility, which increased to over 50,000 sites showing increased accessibility and 8000 showing decreased accessibility by 4 hr (*Figure 1C*). These data are consistent with previous reports of overall increased accessibility upon CHD4/dMi2 depletion by shRNA in mouse ES cells or *Drosophila* S2 cells (*de Dieuleveult et al., 2016*; *Moshkin et al., 2012*).

Differentially accessible regions increasing in accessibility upon CHD4 depletion exhibited low accessibility and low overall enrichment for specific histone modifications associated with active chromatin in undepleted cells (H3K27Ac, H3K4Me1, and H3K4Me3), but not H3K27Me3 (*Figure 1D, E*). As a class, these sites predominantly mapped to inactive enhancers and repetitive elements (*Figure 1F*). They had very low average enrichment for CHD4 or MBD3 (*Figure 1G*) and would likely not pass the threshold to be counted as 'Peaks' in many ChIP-seq or Cut&Run datasets. Low enrichment for active chromatin marks and NuRD components, as well as low but detectable accessibility, indicates that these sites predominantly represent silent and cryptic regulatory sequences. Upon CHD4 loss, accessibility at these sites increased two- to threefold at 60 and 240 min of depletion (*Figure 1D*), indicating that the low-level CHD4 enrichment at these sites may nevertheless be functionally relevant.

Sites showing decreased accessibility without CHD4 were highly accessible in undepleted cells, enriched for marks of active chromatin such as H3K4Me1 and H3K27Ac, and to a lesser extent H3K4Me3 (*Figure 1D, E*), and often overlapped with known enhancers and promoters (*Figure 1F*). These sites also showed high enrichment for CHD4 and MBD3 in the undepleted state (*Figure 1G*). Upon CHD4 depletion, accessibility at these sites was reduced by less than twofold on average (*Figure 1D*). We and others have previously shown that NuRD acts at active enhancers (*Basu et al., 2023*; *Bornelöv et al., 2018*; *de Dieuleveult et al., 2016*) and, consistent with these findings, CHD4 depletion caused an average decrease in accessibility at active enhancers (*Figure 1H*). We therefore conclude that CHD4 activity maintains closed chromatin generally at inactive regulatory sequences but also contributes to the maintenance of highly accessible chromatin at active sites.

### CHD4 activity maintains expression of active genes while reducing transcriptional noise

Significant changes in gene expression were first detected in nascent and total mRNA 1–2 hr after CHD4 depletion (*Figure 2A, B*). Loss of CHD4 resulted in an approximately 2:1 ratio of increased to

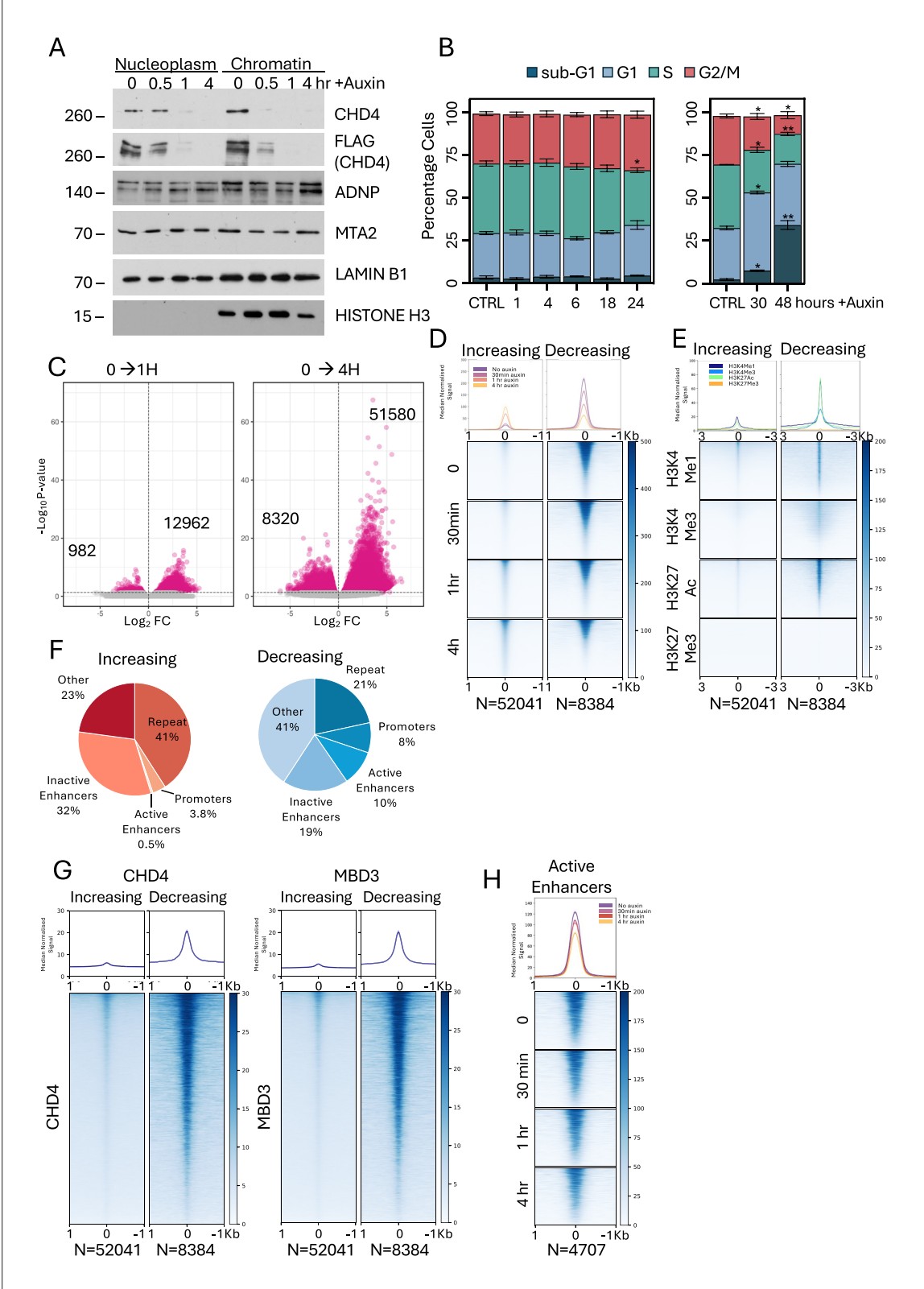

**Figure 1.** The impact of CHD4 depletion on chromatin accessibility. (**A**) Western blots of nuclear soluble (nucleoplasm) and chromatin fractions across CHD4 depletion probed with antibodies directed against indicated proteins. Times in hours of Auxin addition are indicated across the top. Lamin B1 and Histone H3 act as loading controls. Position of relevant size marker indicated at left in KDa. (**B**) Cell cycle analysis across CHD4 depletion time course. Hours post Auxin addition are indicated at the bottom, 'CTRL' indicates DMSO control. Data represent an average of three replicates. Asterisks

*Figure 1 continued on next page*

*Figure 1 continued*

indicate significant differences from CTRL using a mixed-effects model with Dunnett's multiple comparisons corrections. Error bars show standard error of the mean. *p < 0.05; **p < 0.01. Representative images of cells and cell cycle profiles are provided in *Figure 1—figure supplement 1*. (**C**) Volcano plots of differentially accessible ATAC-seq peaks between 0 and 1 hr of Auxin addition (left) or 0 and 4 hr of Auxin addition (right). Magenta spots indicate statistically significant differences (FDR >0.05). Numbers of peaks that decreased or increased significantly are indicated on the plots. (**D**) Heatmaps of ATAC-seq signal for all regions displaying increased accessibility (*N* = 52,041) or decreased accessibility (*N* = 8384) at any time across the CHD4 depletion time course are displayed for each time point. (**E**) Heatmaps of Cut&Run data for indicated histone modifications at sites increasing or decreasing in accessibility (as in panel **D**) at indicated times after CHD4 depletion. (**F**) Percentages of sites increasing in accessibility (top, blue) or decreasing in accessibility upon CHD4 depletion (bottom, red) which localise to indicated genomic features. Active enhancers are defined as having H3K4Me1 and K3K27Ac but not K3K4Me3, and inactive enhancers as having H3K4Me1 but not H3K4Me3 or H3K27Ac. (**G**) Heatmaps of CHD4 and MBD3 Cut&Run data at upDARs and downDARs in 2iL conditions. (**H**) Heatmaps of ATAC-seq signal at active enhancers (*N* = 4707) across the CHD4 depletion time course. Median curves in graphs in **D, E, G**, and **H** are plotted with standard error of the mean in lighter shading.

The online version of this article includes the following source data and figure supplement(s) for figure 1:

**Source data 1.** PDF file containing original western blots for *Figure 1A*, indicating the relevant bands and conditions.

**Source data 2.** Original files for western blots displayed in *Figure 1A*.

**Figure supplement 1.** Cell cycle arrest and cell death following CHD4 depletion.

decreased gene expression from 1 hr onwards (*Figure 2C*), consistent with CHD4 activity able to both facilitate and limit transcription, with most genes changing by twofold or less (*Figure 2D*). GO terms associated with activated genes indicate various tissue-specific functions, consistent with the general noise reduction activity described for NuRD (*Burgold et al., 2019*; *Montibus et al., 2024*; *Ragheb et al., 2020*). Downregulated genes, in contrast, are associated with cellular maintenance and early development, consistent with these genes being normally highly expressed in ES cells (*Figure 2E*).

To assess whether the observed changes in chromatin accessibility were linked to changes in gene expression, we plotted the distance between differentially accessible regions and the annotated TSS of genes found to be misexpressed up to 4 hr after CHD4 depletion. Sites increasing in accessibility upon CHD4 depletion tended to be located far from genes repressed by CHD4 (*Figure 3A*). This makes it unlikely that CHD4 directly silences gene expression by maintaining condensed chromatin at inactive promoters widely, although it could be acting to prevent activation of distal enhancers. In contrast, a notable number of sites losing accessibility are located within 1 kb of the TSS of misregulated genes (*Figure 3A*). This indicates that CHD4 acts at some highly active promoters to maintain both chromatin accessibility and transcriptional fidelity.

If loss of CHD4 resulted in activation of normally silent or cryptic enhancers, we would expect that they should show increases in enhancer-associated histone modifications and binding of TFs. As expected, Cut&Tag for H3K4Me1 and H3K27Ac showed increasing enrichment at CHD4-condensed sites 4 hr after CHD4 depletion, which did not increase further by 24 hr of depletion (*Figure 3B*). Although these sites were largely unbound by NANOG or SOX2 in the presence of CHD4, within 1 hr of CHD4 depletion, they became extensively bound by both TFs (*Figure 3C*). Consistently, thousands of new NANOG and SOX2 peaks were detected across the genome after CHD4 depletion (*Figure 3D*). For example, the 25 kb region upstream of the *Eomes* gene contains multiple CHD4-condensed sites (*Figure 3E*). Those labelled 1, 2 and 3 show very little, if any, accessibility or TF binding, and very low H3K27 acetylation and H3K4 monomethylation in undepleted cells. All three of these show a gain in accessibility, TF binding and both histone modifications upon CHD4 depletion. Those sites labelled 4 and 5 both show some accessibility, TF binding and enrichment for active histone modifications in undepleted cells. Nevertheless, these sites also show an increase in all of these features upon CHD4 depletion. In contrast, the nearby *Eomes* promoter (P) shows little or no change in accessibility, TF binding or enrichment for H3K27Ac or H3K4Me1 across the CHD4 depletion time course.

We propose that low-level association of CHD4 and NuRD across euchromatin restricts chromatin accessibility, preventing TFs from binding to consensus sequence motifs. It is also possible, however, that accessibility increases are a consequence of more stable TF binding. Failure to maintain this level of chromatin inaccessibility results in activation of inactive and/or cryptic enhancers, which can stimulate inappropriate gene expression.

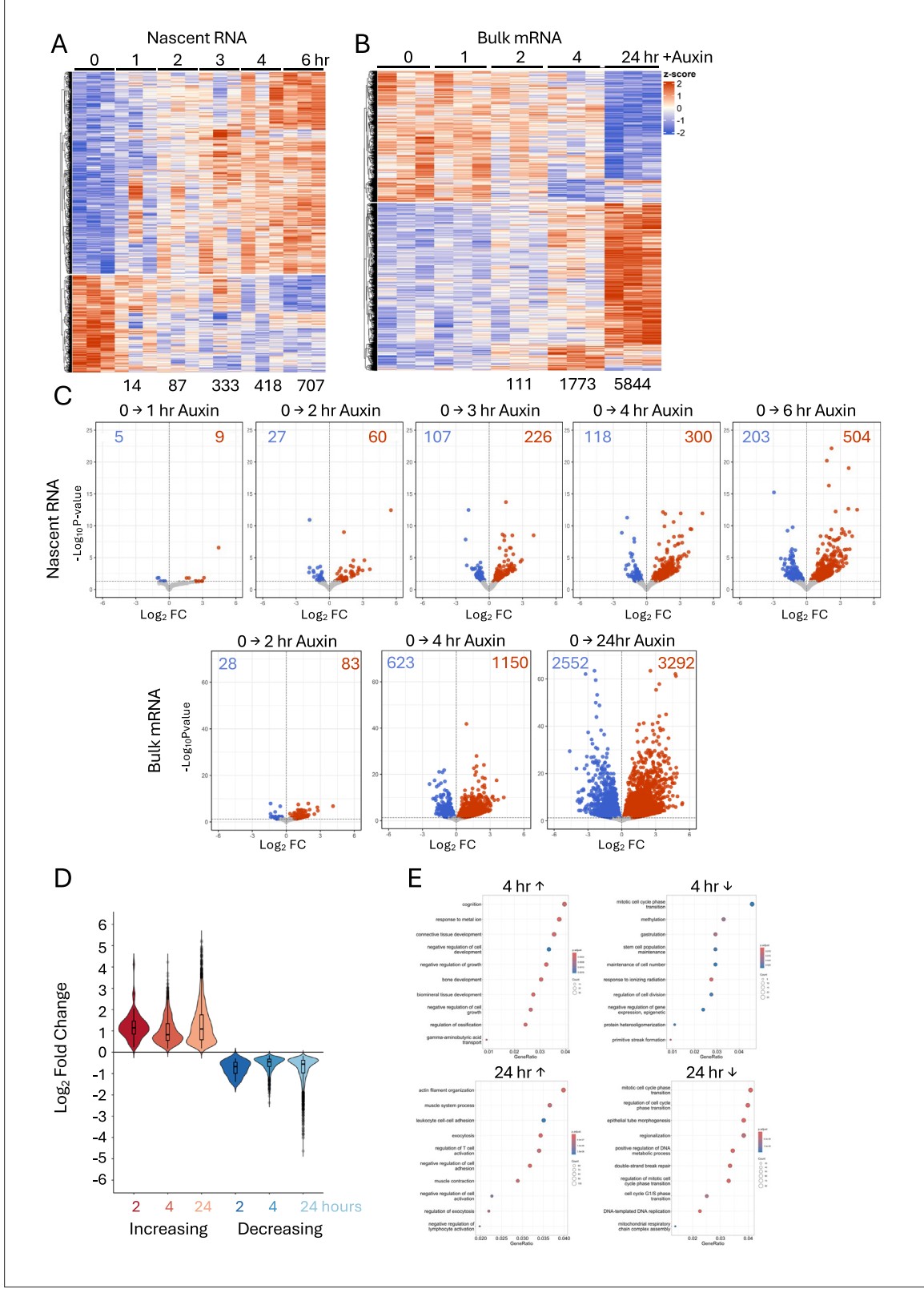

**Figure 2.** CHD4 acutely regulates gene expression. Heatmaps from nascent RNA-seq (**A**) or bulk RNA-seq (**B**) of genes showing significant ($p_{adj} < 0.05$) changes in expression at any point during the CHD4 depletion time course. Heatmaps display z-scores, meaning expression for each gene has been centred and scaled across the entire time course. (**C**) Volcano plots showing significant gene expression changes at indicated time points in nascent RNA-seq (top) and bulk RNA-seq (bottom). Genes increasing upon CHD4 depletion are shown in red, and those decreasing are shown in blue. The

*Figure 2 continued on next page*

*Figure 2 continued*

number of significantly misexpressed genes at each time point is indicated in the figure. (**D**) Violin plots showing the average $\log_2$ fold change of significant upregulated (red) and downregulated (blue) genes during the CHD4 depletion time course. (**E**) Gene ontology (GO) enrichment analysis of genes increased or decreased after 4 or 24 hr of CHD4 depletion. The top 10 biological processes are shown for each category, based on smallest adjusted p-value.

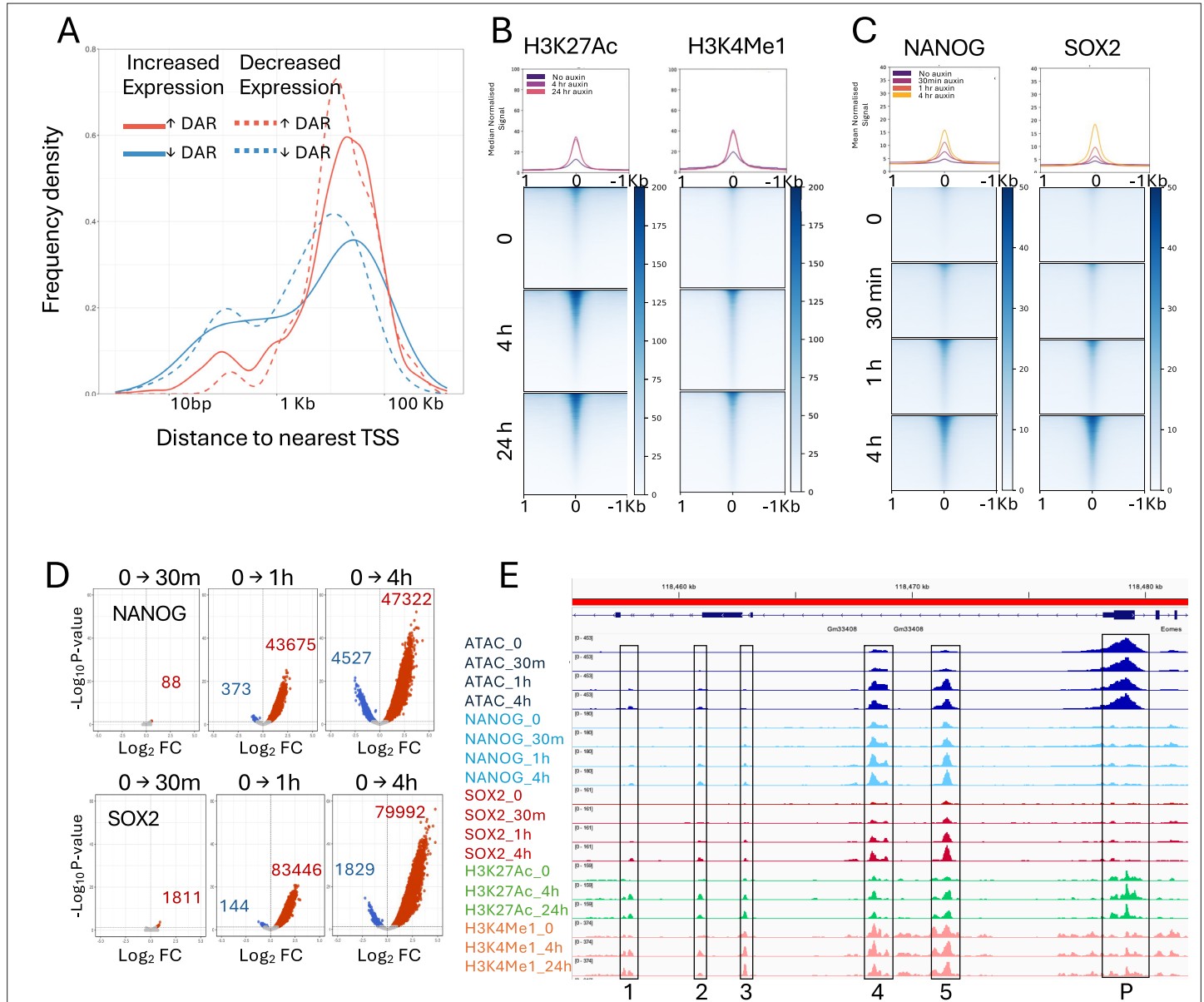

**Figure 3.** Chromatin opening upon CHD4 depletion. (**A**) Frequency density distribution of the distance of increasing differentially accessible regions ('↑ DAR', red lines) and decreasing regions ('↓ DAR', blue lines) to the TSS of genes showing increased (solid lines) or decreased (dotted lines) expression within 4 hr of CHD4 depletion. (**B**) Heatmaps of Cut&Tag signal for H3K27Ac and H3K4Me1 at sites increasing in accessibility ($N$ = 52,041) at indicated times of CHD4 depletion. (**C**) Heatmaps of NANOG and SOX2 Cut&Run signal at increasing accessibility sites at indicated times of CHD4 depletion. Median curves in **B and C** are plotted with standard error of the mean in lighter shading. (**D**) Pairwise comparisons of called peaks of binding for NANOG (top) and SOX2 (bottom) between undepleted cells (0 hr) and 30 min, 1 hr, or 4 hr of CHD4 depletion. Significantly changed (FDR >0.05) binding sites are shown in blue when $\log_2$ FC >0 and red when $\log_2$ FC <0. (**E**) IGV screenshot of the upstream region of the mouse *Eomes* locus displaying ATAC-seq, Cut&Run, and Cut&Tag data as indicated at left. Boxed regions labelled 1–5 are CHD4-condensed sites, while the box labelled P corresponds to the *Eomes* promoter.

## CHD4 and SALL4 maintain chromatin inaccessibility largely independently

The SALL4 protein is a very abundant TF in mouse ES cells, which has long been known to interact with the NuRD complex (*Kloet et al., 2015*; *Lauberth and Rauchman, 2006*; *Lu et al., 2009*; *Miller et al., 2016*). SALL4 preferentially binds to A/T-rich DNA genome-wide, where it is proposed to exert a general repression function through NuRD recruitment (*Kong et al., 2021*; *Pantier et al., 2021*; *Ru et al., 2022*; *Watson et al., 2023*). If this model is correct, we would expect some proportion of the CHD4-condensed sites to be dependent upon both CHD4 and SALL4 to remain inaccessible. To test this model, we assessed chromatin accessibility after dTAG-mediated SALL4 depletion. SALL4 is partially redundant with SALL1 in ES cells, but only SALL4 is required for early mammalian development (*Miller et al., 2016*; *Nishinakamura et al., 2001*). We therefore created *Sall1$^{-/-}$Sall4$^{FKBP/-}$* ES cells in which one endogenous *Sall4* allele was mutated and the remaining allele was targeted to express a SALL4-FKBP fusion protein.

Acute depletion of SALL4-FKBP resulted in rapid and extensive changes in chromatin accessibility as measured using ATAC-seq. Over 18,000 sites showed increased accessibility within 1 hr of SALL4 depletion in *Sall1$^{(-/-)}$* ES cells (*Figure 4A, B*). These sites showed moderate enrichment for SALL4 in undepleted cells, though less than that seen at active enhancers (*Figure 4C*). When we compared these sites to those showing CHD4-dependent chromatin condensation, we found that only 4559 sites of the more than 52,000 CHD4-dependent sites also show dependency upon SALL4 to maintain chromatin inaccessibility (*Figure 4D–F*). This means that only 17.2% of SALL4-dependent sites also rely on CHD4 to prevent chromatin opening. Moreover, over 47,000 CHD4-dependent sites show neither SALL4 binding nor SALL4 dependence to remain inaccessible (*Figure 4D-G*). Similarly, over 20,000 SALL4-dependent sites show no change in accessibility upon CHD4 depletion, despite showing similar levels of enrichment for NuRD components as those dependent upon both SALL4 and CHD4 (*Figure 4E, H*). Consistent with the proposed preference of SALL4 for binding to A/T-rich DNA, SALL4-unique regions showed enrichment for SALL4 binding and a high A/T content (53.9%) (*Figure 4D*). CHD4-unique regions showed little if any SALL4 enrichment and a much lower A/T content (44.2%), while those sites requiring both SALL4 and CHD4 to remain inaccessible showed enrichment for NuRD components as well as SALL4 and displayed intermediate A/T content (48.2%). Together, these data show that both CHD4 and SALL4 broadly restrict chromatin accessibility across the genome, but that their activities overlap at only a relatively small proportion of sites.

## NuRD limits TF binding to chromatin

Sites decreasing in accessibility upon CHD4 depletion showed an increase in enrichment for both NANOG and SOX2 after 30 min, even though these sites were, on average, losing accessibility and losing enrichment for the active chromatin marks H3K4Me1 and H3K27Ac at that time (*Figure 5A, B*). Active enhancers similarly showed an increase in NANOG and SOX2 enrichment from 30 min of CHD4 depletion (*Figure 5C*). An example locus is shown in *Figure 5D*, where a cluster of enhancers located 50–70 kb downstream of the *Klf4* gene contains several high accessibility sites. Those located at 68, 57, and 55 kb downstream of *Klf4* (labelled 68, 57, and 55, respectively, in *Figure 5D*) all show a loss of ATAC-seq signal but a gain in NANOG and, to a lesser extent, SOX2 enrichment between 0 and 30 min of CHD4 depletion. NANOG enrichment then decreases again at 1 and 4 hr while Sox2 enrichment remains relatively constant, even though accessibility continues to decrease.

The simultaneous decrease in chromatin accessibility and increase in TF enrichment after 30 min of CHD4 depletion could indicate that CHD4/NuRD plays a direct role in controlling TF binding to chromatin. To independently assess the impact CHD4 has on TF binding, we used single-molecule imaging in live ES cells to directly measure TF-binding kinetics upon CHD4 depletion. A HALO-Tag cassette was knocked in to the endogenous *Nanog*, *Klf4* or *Sox2* loci in the CHD4-mAID ES cell line to create strains expressing the different C-terminal protein fusions. Single HALO-tagged protein molecules were then labelled with a photoactivatable dye (PA-JF646, *Grimm et al., 2016*) and tracked at two distinct temporal regimes where we collected images either every 20 or 500 ms, using double-helix point spread function microscopy as they moved within a 4-µm slice of the nucleus. Recording at a 20-ms time resolution allows the segmentation of trajectories into freely diffusing and chromatin-bound states, and this data can be used to extract the chromatin-binding kinetics of proteins and complexes (*Basu et al., 2023*). Here, we used these data to determine residence times – how long the

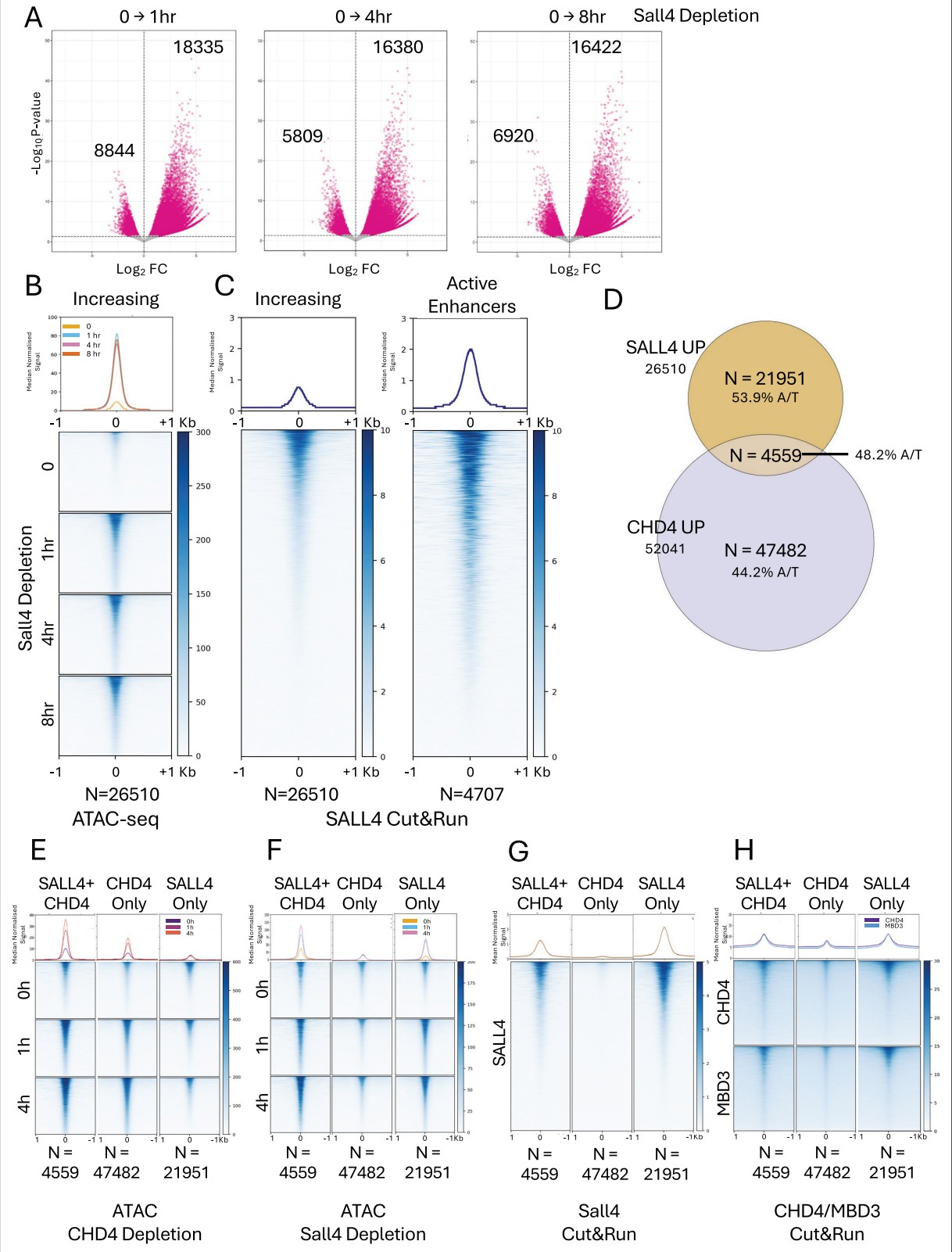

**Figure 4.** CHD4 and SALL4 both restrict chromatin accessibility. (**A**) Volcano plots of differentially accessible ATAC-seq peaks when comparing 1, 4, or 24 hr of SALL4 depletion with those seen in undepleted cells (0). Magenta spots indicate statistically significant differences (FDR >0.05). Numbers of peaks that decreased or increased significantly are indicated on the plots. (**B**) Heatmaps of ATAC-seq signal for all regions displaying increased accessibility (N = 26,510) across the SALL4 depletion time course are displayed for each time point. (**C**) Heatmap of SALL4 Cut&Run signal in

*Figure 4 continued on next page*

*Figure 4 continued*

undepleted embryonic stem (ES) cells (taken from **Ru et al., 2022**) at all regions displaying increased accessibility across the SALL4 depletion time course (left, N = 26,510) or at active enhancers (right, N = 4707). (**D**) Overlap of sites showing increased accessibility upon SALL4 depletion with those increasing upon CHD4 depletion (upDARs). The % A/T base composition of the different categories of sites is indicated. Heatmaps of ATAC-seq signal at sites increasing upon either SALL4 or CHD4 depletion (SALL4 + CHD4), sites increasing upon CHD4 depletion but not upon SALL4 depletion (CHD4 only), or sites increasing upon SALL4 depletion but not upon CHD4 depletion (SALL4 only) plotted at indicated time points of CHD4 depletion (**E**) and SALL4 depletion (**F**). Cut&Run signal for SALL4 (**G**) or for CHD4 and MBD3 (**H**) in undepleted ES cells at the three different classes of sites. Median curves in **B, C**, and **E–H** are plotted with standard error of the mean in lighter shading.

molecules remain bound to chromatin. The proportion of molecules that remain bound to chromatin after increasing lengths of time was then plotted before and after CHD4 depletion (for 1 hr) and the results were fitted with a single exponential decay to determine the apparent dissociation rate, $k_{off}$ (**Figure 5E**). NANOG-HALO, SOX2-HALO, and KLF4-HALO all showed a pronounced decrease in apparent $k_{off}$ after CHD4 depletion (**Figure 5F–H**), consistent with CHD4 actively limiting TF residence times on chromatin.

To determine whether it is CHD4 by itself or the intact NuRD complex that can remodel TF binding, we similarly imaged NANOG-HALO and KLF4-HALO in MBD3-depletable ES cells after disruption of the NuRD complex (**Figure 5I**). These cells were also null for *Mbd2*, which encodes a protein displaying partial functional redundancy with MBD3 but is dispensable in ES cells (**Hendrich et al., 2001**). As CHD4 is largely responsible for the binding of NuRD to its target enhancers, removal of MBD3 enables disassembly of NuRD, but does not prevent CHD4 binding to chromatin (**Basu et al., 2023**; **Bornelöv et al., 2018**; **Zhang et al., 2016**). The nucleosome remodelling activity of CHD4 is, however, greatly reduced outside of NuRD (**Bornelöv et al., 2018**). We therefore reasoned that if it is intact NuRD that is required (as opposed to just CHD4), depletion of MBD3 should also limit TF residence times. Indeed, we detected a significant decrease in the apparent $k_{off}$ of stably bound NANOG-Halo and KLF4-Halo after 1 hr of MBD3 depletion (**Figure 5J, K**). We therefore conclude that intact NuRD (and not CHD4 by itself) actively limits TF residence times on chromatin, while at the same time contributing to the maintenance of chromatin accessibility at active regulatory regions.

## NuRD activity influences active and inactive regulatory elements differently

We next asked how CHD4 activity could have different impacts on accessibility at different kinds of sequences. Information about the structure of accessible regions can be obtained from ATAC-seq data by quantitating recovered fragments of all sizes and determining the frequency and location of recovered Tn5 integration sites using Vplots (**Henikoff et al., 2011**; **Schep et al., 2015**; **Serizay and Ahringer, 2021**; **Figure 6—figure supplement 1**). Applying this analysis at CHD4-condensed sites shows an increase in both Tn5 integrations and in reads corresponding to the nucleosome-free region (NFR) after 30 min of CHD4 depletion (**Figure 6A, B**). After 60 min, when CHD4 is almost completely depleted, there is an increase in accessibility generally across the entire site, both of short reads within the NFR but also longer reads (200–300 bp) extending from the NFR outwards (**Figure 6B**). We conclude that NuRD is acting to restrict accessibility of both the NFR and the flanking nucleosomal DNA (see **Figure 6—figure supplement 1**) at these sites, consistent with the demonstrated role of CHD4/NuRD in maintaining the density of intact nucleosomes (**Bornelöv et al., 2018**; **de Dieuleveult et al., 2016**; **Moshkin et al., 2012**).

The example of the *Eomes* locus (**Figure 3F**) illustrates that some CHD4-condensed sites show a degree of accessibility in the presence of CHD4, while others show little if any accessibility. To better understand how CHD4 prevents chromatin opening at these two different kinds of sites, we constructed Tn5 integration plots and Vplots from the 1000 CHD4-condensed sites showing the least accessibility in undepleted conditions and from the 1000 showing the most accessibility (**Figure 6C–F**). Sites showing little or no accessibility in undepleted cells initially show very few reads less than 100 bp. The Vplots also show reads of 200–300 bp distributed across the regions, again indicative of generally inaccessible chromatin. Loss of CHD4 results in the formation and progressive expansion of an NFR, as well as an increase in longer reads with one end located within the NFR (**Figure 6C, D**). The Tn5 integration plot does not show a general widening of the NFR over time, indicating that CHD4 function is largely focussed on maintaining nucleosome density within a 100- to 200-bp region (**Figure 6C**).

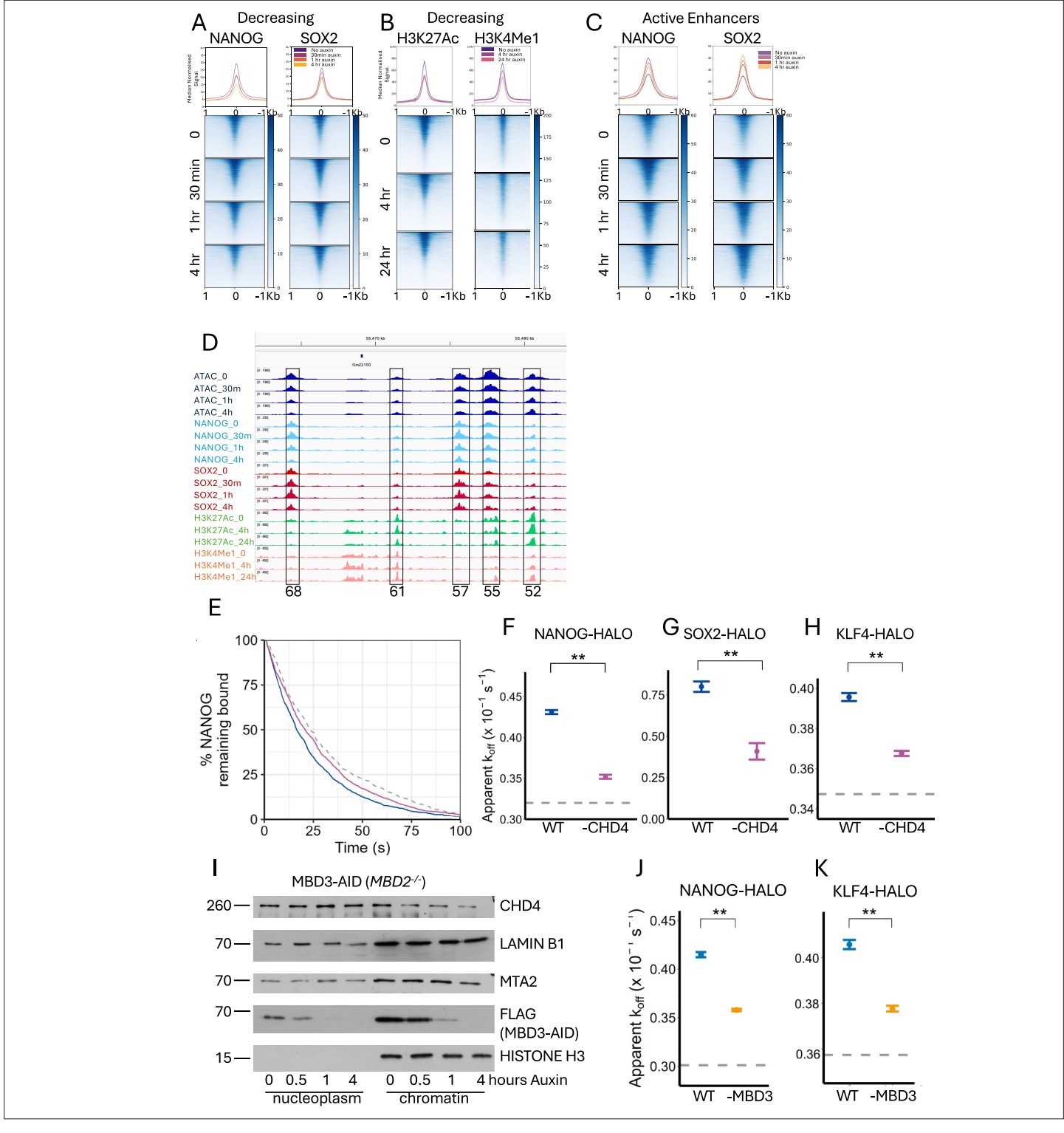

**Figure 5.** Nucleosome Remodelling and Deacetylation (NuRD) regulates NANOG and SOX2 binding to active sites. (**A**) Heatmaps of NANOG and SOX2 Cut&Run signal at indicated times of CHD4 depletion across sites decreasing in accessibility ($N$ = 8384). (**B**) Heatmaps of H3K27Ac and K3H4Me1 Cut&Tag signal at indicated times of CHD4 depletion across decreasing accessibility sites. (**C**) Heatmaps of Cut&Run signal for NANOG and SOX2 across active enhancers ($N$ = 4707) at indicated times of CHD4 depletion. Median curves in **A–C** are plotted with standard error of the mean in lighter shading. (**D**) IGV screenshot of the enhancer cluster downstream of the *Klf4* gene displaying ATAC-seq, Cut&Run, and Cut&Tag data as indicated at the left. Boxed regions are labelled with the distance in kb from the annotated *Klf4* transcription start site. (**E**) Fluorescence survival curves of chromatin-bound NANOG-HALO molecules in 2iL (blue line) or after 1 hr of CHD4 depletion (purple line). The grey dotted line represents the fluorescence

*Figure 5 continued on next page*

*Figure 5 continued*

survival curve for molecules in a fixed-cell control imaged under identical conditions. (**F**) Apparent dissociation rates ($k_{off}$) of chromatin-bound NANOG molecules calculated through fitting a single exponential decay model to the survival curves in panel **E**. Error bars represent 95% confidence intervals for each fit applied to data taken from three independent experiments. The horizontal dashed line represents the upper 95% confidence limit for a fixed-cell control. ** indicates that 99% confidence intervals do not overlap, that is p < 0.01. As in **F** but for SOX2 (**G**) and KLF4 (**H**). The fixed-cell control was not imaged in the SOX2 experiments in panel **F**. (**I**) Western blots of nuclear soluble (nucleoplasm) and chromatin fractions across MBD3 depletion probed with antibodies directed against the indicated proteins. Times in hours of Auxin addition are indicated across the top. Lamin B1 and Histone H3 act as loading controls. Position of relevant size marker indicated at left in KDa. (**J, K**) Apparent dissociation rates ($k_{off}$) for NANOG-HALO (**I**) and KLF4-HALO (**J**) before and after 60 min of MBD3 depletion. For the calculation of $k_{off}$, the trajectories were pooled from four replicates of each time point obtained over 2 days.

The online version of this article includes the following source data for figure 5:

**Source data 1.** PDF file containing original western blots for *Figure 5I*, indicating the relevant bands and conditions.

**Source data 2.** Original files for western blots displayed in *Figure 5I*.

Sites at which CHD4 activity restricts accessibility at already accessible chromatin showed a general increase in both small and longer reads as CHD4 was depleted, and a broadening of the existing NFR (*Figure 6E, F*). Together, these data are consistent with CHD4/NuRD maintaining chromatin compaction at inactive or low activity enhancers by maintaining nucleosome density.

The sites at which CHD4 activity maintains accessibility displayed a prominent NFR as well as high density of longer reads in undepleted cells (*Figure 6G, H*). CHD4 depletion induced a general decrease in Tn5 integrations and Vplot signal intensity across these sites, consistent with CHD4 acting to generally maintain accessibility.

We next asked why CHD4 promoted accessibility at highly accessible active sites (e.g. *Figure 6G, H*) but reduced accessibility at slightly less accessible, inactive sites (e.g. *Figure 6E, F*). We took advantage of ES cells in which a tamoxifen-inducible MBD3b is expressed in an otherwise *Mbd3*-null ES cell line, allowing us to restore NuRD activity to cells upon tamoxifen addition (*Bornelöv et al., 2018*; *Reynolds et al., 2012*). We had previously subjected these cells to micrococcal nuclease (MNase) sequencing at different time points after MBD3 induction to show that restoration of NuRD activity caused increased density of intact nucleosomes across sites of active transcription (*Bornelöv et al., 2018*). Unlike ATAC-seq, sequencing of MNase-treated DNA results in increased reads at sites protected from digestion by bound proteins, that is less accessible regions, and a decrease in signal at locations where the DNA is easily accessible and therefore digested by the MNase. In that study, we used an MNase concentration at which DNA associated with intact nucleosomes is protected from MNase digestion. Lower MNase concentrations, however, will recover DNA protected by other chromatin proteins as well as by partial (fragile) nucleosomes (*Kubik et al., 2015*; *Xi et al., 2011*).

We therefore subjected cells undergoing the time course of NuRD reformation to lower MNase concentrations prior to sequencing. Traditional MNase sequencing shows that NuRD increases the density of intact nucleosomes at both classes of accessible sites, as indicated by an increase in signal from the red line (NuRD-deficient) to the black line (24 hr NuRD restored) in the top panels of *Figure 6I*. This effect is more pronounced at the most accessible inactive sites ('UP Most Accessible', black line, *Figure 6I*), occurring across a broader area than at fully active sites ('All Decreasing', black line). The lower MNase concentration, by contrast (*Figure 6I*, lower panels) produces a peak of MNase resistance in the absence of NuRD activity at both classes of sites (red lines), indicating an accumulation of structures which are digested by the higher MNase concentration, such as fragile nucleosomes. Restoration of NuRD activity clears the majority of this signal at the most accessible inactive sites, but only moderately reduces their abundance at fully active sites (*Figure 6I*, lower panels). These data indicate that at highly active regions, NuRD acts to limit the abundance of fragile nucleosomes and other proteins, while marginally increasing the density of intact nucleosomes, which has an overall result of maintaining these sites in a fully active state. It does not completely remove the fragile nucleosomes from these sites, which are presumably being created by the activity of other chromatin remodellers (*Klein et al., 2023*; *Nocente et al., 2024*). In contrast, at accessible but inactive sites, it prevents the accumulation of fragile nucleosomes which would otherwise cause further opening and enable activation of inactive regulatory elements. This means that the activity of CHD4 does not differ at the two classes of sites, but it is the presence of other remodellers producing fragile nucleosomes which dictates whether CHD4 activity acts to maintain or prevent accessibility (*Figure 7*).

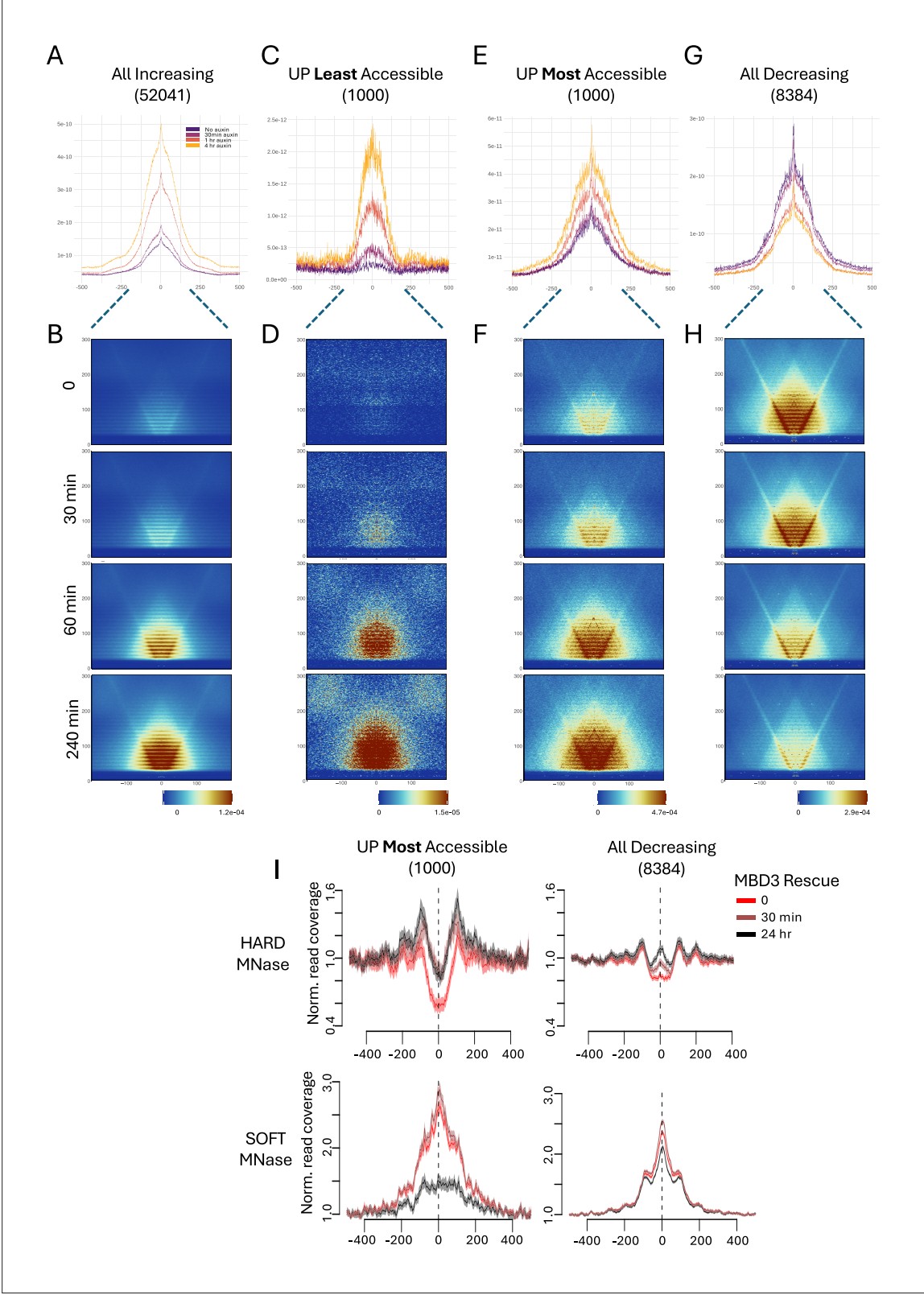

**Figure 6.** CHD4 controls accessibility differently at different classes of sites. (**A, C, E, G**) Tn5 integration frequency was determined from ATAC-seq data and plotted across indicated sites across the CHD4 depletion time course. The number of sites is shown in parentheses. (**B, D, F, H**) Vplots across sites indicated above (corresponding to the Tn5 integration plots) at indicated times of CHD4 depletion. See *Figure 6—figure supplement 1* for a Vplot schematic. (**I**) MNase-seq data collected 0 (red), 30 min (dark red), or 24 hr (black) after Nucleosome Remodelling and Deacetylation (NuRD) reformation

*Figure 6 continued on next page*

Figure 6 continued

in *Mbd3*⁻/⁻ embryonic stem (ES) cells are plotted across indicated sites. Top graphs show results from 'Hard' MNase treatment while bottom graphs show 'Soft' MNase treatment (see text). The *y*-axis shows normalised read coverages, while the *x*-axis shows distance in base pairs from the centre of the feature. Curves show mean and standard error from three biological replicates.

The online version of this article includes the following figure supplement(s) for figure 6:

**Figure supplement 1.** Schematic diagram of Vplots.

Together, these data show that CHD4 shows two modes of binding on chromatin: in addition to the high enrichment seen at active enhancers and promoters, it also shows low-level binding at silent and/or cryptic regulatory elements across euchromatin. At the former class of sites, it acts to limit the residence times of chromatin-associated proteins to maintain highly accessible chromatin. This maintenance activity of CHD4/NuRD is necessary for the full activity of these elements to direct appropriate transcription of target genes. At the latter class of sites, we propose it maintains chromatin in an inaccessible state, preventing accumulation of fragile nucleosomes and preventing TFs from recognising and stably binding their cognate motifs and thereby preventing spurious activation of these elements.

## Discussion

Assessing CHD4 function by traditional knockout experiments is difficult as ES cells without CHD4 undergo cell cycle arrest and begin to apoptose after 24 hr (*Figure 1B*). Even with siRNA or inducible deletion, cells in a population will lose CHD4 at different times and, in the case of siRNA, to different extents, making primary function difficult to ascertain. Genetic deletion in mouse ES cells has been used extensively to define protein function in cells and in intact mice. In such an experiment, the genetic lesion will often be induced in a single cell, and a clone of cells will be grown out from that single cell. While this is an extremely powerful method for defining gene function, there is usually a very large number of cell divisions occurring and considerable time passing between loss of the gene and assessment of phenotype, making it of limited use when studying proteins required for the viability of ES cells. Resulting phenotypes may be due directly to the absence of protein function but may also reflect adaptations made by cells selected to proliferate in the absence of that protein. The advent of inducible depletion methods (summarised in *de Wit and Nora, 2023*) has meant we can now infer protein function by assessing the immediate molecular consequences of loss of a given protein activity in the very short term. Combined with genome-wide profiling, this has allowed us to identify both a novel TF remodelling activity for CHD4 and to define how this combines with the protein's nucleosome remodelling activity to exert different functions at different classes of sites.

CHD4 is an abundant chromatin remodeller with a well-defined function in moving intact nucleosomes. Here we show that CHD4 can also actively limit the residence times of TFs on chromatin. The combination of these two activities has different consequences at different kinds of sequences. At highly accessible, active sites (*Figure 7A*), CHD4 enrichment is high and it acts to limit the binding of TFs and fragile nucleosomes to regulatory regions, allowing for full accessibility and functionality of these regulatory elements. At inactive, less accessible euchromatic sites (*Figure 7B*), CHD4 associates more infrequently, but here it is TF removal activity, combined with maintenance of nucleosome structure, is sufficient to keep accessibility low and to prevent TFs from stably binding recognition motifs. This activity is sufficient to prevent the binding of NANOG and SOX2, the latter being capable of binding nucleosomal templates (*Dodonova et al., 2020*; *Zhu et al., 2018*). By combining these two activities, CHD4 functions to prevent spurious activation of cryptic and/or silent regulatory regions, thereby reducing transcriptional noise, but also acts to facilitate the activity of highly accessible regulatory regions, allowing cells to accurately and rapidly respond to differentiation cues (*Figure 7*).

Many TFs recognise and bind to specific sequence motifs in DNA. A TF recognising a 7-base pair sequence should, on average, have more than 150,000 different potential binding sites in a mammalian genome. Yet ChIP-seq and Cut&Run studies have found most TFs associate with orders of magnitude fewer sites in any one cell type (*Lambert et al., 2018*; *Srivastava and Mahony, 2020*). Most TFs bind preferentially or exclusively in accessible chromatin, so it is believed that differences in chromatin accessibility, or nucleosome positioning across sites, determine which subset of the potential consensus DNA-binding motifs are available for TF binding at any given time or cell type (*Workman and Buchman, 1993*; *Zhu et al., 2018*).

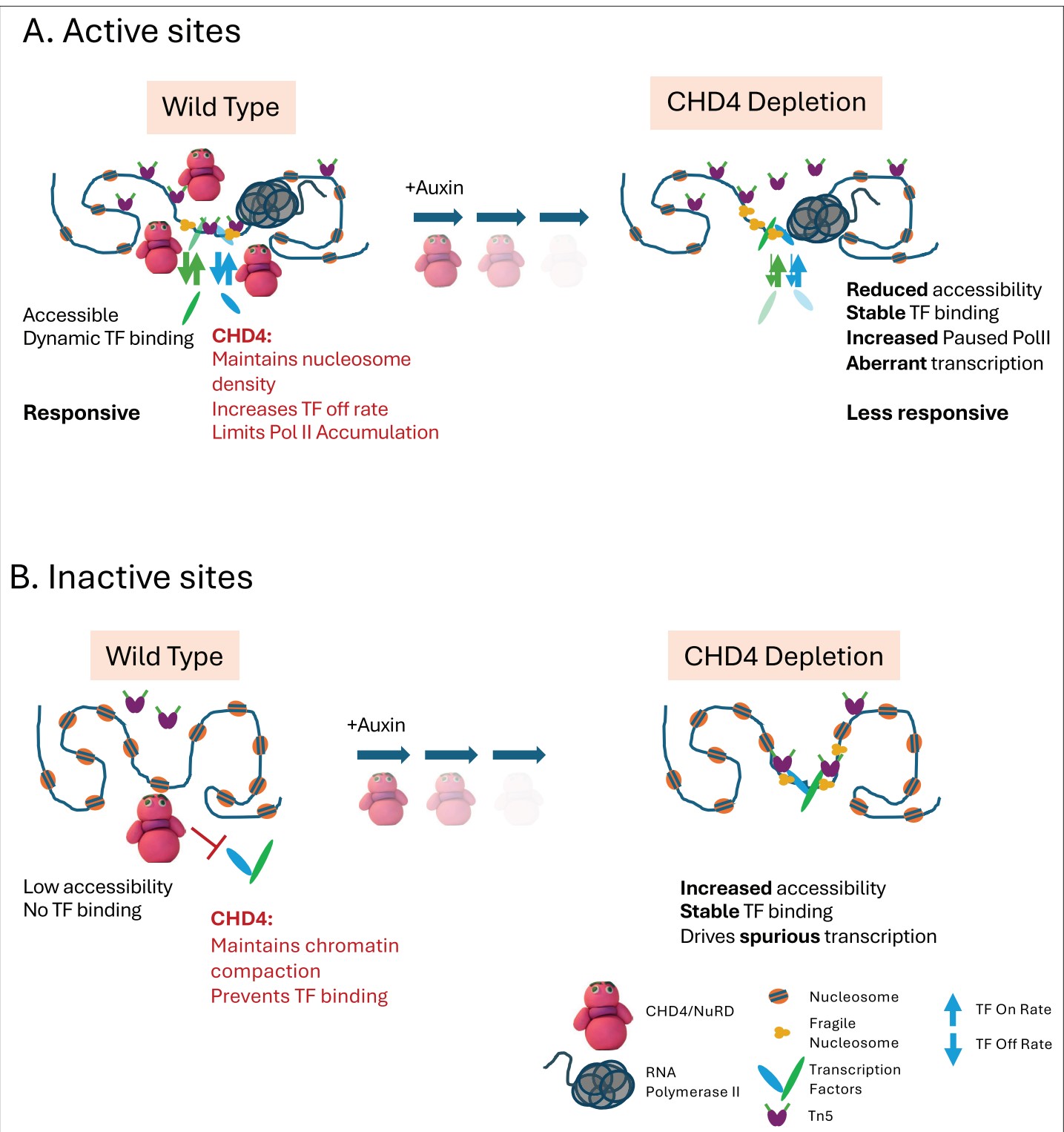

**Figure 7.** Model of CHD4 function. (**A**) Highly accessible sites. In undepleted cells, these sites are extensively bound by CHD4/NuRD, where it acts to promote the off rate of transcription factors (TFs) to promote accessibility. Tn5 is able to access the central nucleosome-free region (NFR) but also can integrate into the flanking nucleosomal DNA. After CHD4 depletion, the on rate for TFs does not change, but the off rate is now much reduced, resulting in increased TF binding. The sites become less accessible to Tn5, such that although it can still access the hypersensitive site within the NFR, there are fewer integrations extending outwards. These regulatory regions cannot quickly respond to receipt of external signals. (**B**) Model of CHD4 function at inaccessible, silent enhancers. In undepleted conditions, there is low CHD4 enrichment at these sites. Here, CHD4 acts to prevent binding of TFs and maintain low accessibility, such that Tn5 cannot frequently access the DNA. After CHD4 depletion, the locus becomes more accessible, and TFs can stably bind. This leads to spurious activation of distal promoters and an increase in transcriptional noise.

Here, we identify CHD4 as one factor which acts to prevent TFs, even so-called Pioneer Factors, from binding thousands of sites in inaccessible chromatin. Consistent with this model, CHD4 was found to prevent binding of the GATA3 TF to inappropriate sites in breast cancer cells undergoing mesenchymal-to-epithelial transition (*Saotome et al., 2024*). We propose that CHD4's chromatin remodelling activity keeps the sites inaccessible, so TF binding is not favoured (i.e. $k_{on}$ is low). The ability of CHD4 to remove TFs from chromatin means that even if they are capable of binding to a nucleosomal substrate, CHD4 promotes their dissociation from chromatin so they are quickly removed (*Figure 7B*). At highly accessible chromatin, CHD4 similarly limits TF residence times by increasing $k_{off}$; however, accessibility is high, so the $k_{on}$ is also high. We propose that at these sites, CHD4 activity balances the high $k_{on}$ by maintaining a high $k_{off}$, thus facilitating the turnover of bound molecules (*Figure 7A*). Without CHD4, $k_{off}$ decreases while $k_{on}$ remains high, resulting in more stable binding of TFs to chromatin. CHD4 and NuRD have been shown to be important for cells to properly respond to differentiation cues in a wide variety of organisms and contexts (*Ahringer, 2000*; *Bracken et al., 2019*; *Lenz and Brehm, 2023*; *McDonel et al., 2009*; *Signolet and Hendrich, 2015*) and we propose that by facilitating TF turnover and accessibility at promoters and enhancers, CHD4 and NuRD facilitate the ability of regulatory elements to respond to differentiation cues (*Figure 7*).

CHD4's activity to prevent spurious TF binding and chromatin opening occurs at many sites where enrichment of CHD4 by ChIP or Cut&Run is low, but not absent. Despite the low amount of CHD4 detected at these sites, this activity was sufficient to prevent activation of silent and cryptic regulatory regions (*Figure 2*). This low-level enrichment of CHD4 at sites where it prevents activation could explain an observation made during *Drosophila* spermatogenesis, where the CHD4 orthologue dMi2 was found to prevent transcription from cryptic promoters, despite not being detected at these promoters by ChIP-seq assays (*Kim et al., 2017*). A recent study in *S. cerevisiae* found many TFs appear to exert influences over expression of genes without detectable association at that gene and bind many sites where they exert no detectable regulatory function (*Mahendrawada et al., 2025*). While some have argued that one should focus on high confidence binding targets to assess protein function (*Mindel and Barkai, 2025*), in our case, this would have resulted in us ignoring most of the 50,000 sites at which we detect CHD4-dependent chromatin compaction (*Figure 1C*). While we have argued that chromatin changes detected within 30 min are likely to be direct consequences of NuRD manipulations (*Figure 6*), we do acknowledge that it is possible these are indirect consequences of loss or gain of NuRD activity. We agree that the definition of 'targets' can be highly variable between studies, which is why we preferred to assess CHD4 enrichment at subsets of sites where function was identified, irrespective of whether it passed the threshold of being a 'peak'.

SALL4 presents another example of a protein exerting function away from its predominant sites of chromatin enrichment: initial ChIP-seq results indicated that SALL4 was predominantly located at enhancers, whereas more recent Cut&Run data indicates a wider spectrum of sites bound by SALL4 (*Kong et al., 2021*; *Miller et al., 2016*; *Pantier et al., 2021*; *Ru et al., 2022*). We find that SALL4 shows lower enrichment at A/T-rich genomic sites than it does at enhancers, but that it nevertheless has a major impact on chromatin compaction at these genomic A/T-rich sites (*Figure 4*). While many groups, including ours, have generally assumed protein function would be focussed at ChIP-seq 'peaks', we argue here that focusing on protein enrichment levels on chromatin is not necessarily the best way to identify important sites of protein activity.

Together, our data show that CHD4 exerts functions on chromatin beyond its well-described ability to slide intact nucleosomes along the DNA (*Zhang et al., 2016*; *Zhong et al., 2020*). Consistent with this conclusion, a recent paper found that acute depletion of CHD4 resulted in an increase in fragile nucleosomes at enhancers (*Nocente et al., 2024*), while we and others have found that NuRD acts to limit the amount of paused RNA Polymerase II at active promoters (*Bornelöv et al., 2018*; *Pundhir et al., 2023*). It is possible that CHD4 translocates along the DNA within accessible sequences and displaces any bound proteins, such as TFs or fragile nucleosomes, that it encounters. Both TFs and fragile nucleosomes are less tightly bound to DNA than intact nucleosomes, so it is possible CHD4's nucleosome remodelling activity results in their displacement, while still sliding intact nucleosomes. CHD4 could also displace TFs from preferred binding sites, thus making binding less favourable. Such activity might not be necessary in inaccessible chromatin, but at inactive enhancers showing a small amount of accessibility, infrequent binding and translocation of NuRD across the accessible region could remove any proteins which have managed to bind. Whether this is achieved through a similar

ATP-dependent mechanism as is used to slide nucleosomes remains to be determined (*Reid et al., 2024*; *Zhong et al., 2020*). Notably, the yeast remodeller INO80 was found to bind differently to fragile nucleosomes versus intact nucleosomes, possibly indicating different mechanisms of remodelling the two different substrates (*Wu et al., 2023*; *Zhang et al., 2023*).

A recent study showed that the nucleosome remodelling activity of a different remodeller, SMARCA5, is dictated by the density of nucleosomes on the DNA that it encounters: at high-density sites, it maintains that density, while at low density sites, it slides nucleosomes across the DNA to facilitate TF access (*Abdulhay et al., 2023*). We find that CHD4 increases density of intact nucleosomes at sites where it prevents or maintains accessibility (*Figure 6I*, top panels) and also limits TF residence times at both classes of sites. Therefore, unlike SMARCA5, we see no evidence that the activity of CHD4 differs between condensed and accessible chromatin, but rather the consequences of that activity are different at different kinds of sites (*Figure 7*).

The residence times of TFs vary but are generally on the order of a few seconds (*Lu and Lionnet, 2021*). Why might it be important to limit the residence times of TFs? We propose that this could fix an enhancer into a specific state, when one important job of enhancers is to be responsive to changes in stimuli. If no change in status quo is required, then after CHD4 promotes eviction of a particular TF from the enhancer, it can continually re-bind and exert its influence over that enhancer. Should signals change, the enhancer needs to be able to remove TFs corresponding to the old signal to make space for determinants of the new signal. In this model, CHD4/NuRD maintains this fluidity of TF interactions, ensuring enhancers are rapidly able to respond immediately upon receipt of differentiation cues.

## Methods
### Mouse ES cells
Mouse ES cells were cultured in 2i+LIF N2B27 media on gelatin-coated plates as described (*Montibus et al., 2024*; *Mulas et al., 2019*). All cell lines were genotyped and tested for mycoplasma regularly. The CHD4-mAID ES cell line was made in BC8, an F1 hybrid from a C57Black/6 and Mus castaneus cross (40, XY), obtained from Anne Ferguson-Smith (Cambridge) (*Strogantsev et al., 2015*). MBD3-AID ES cells were created in 23AF, a primary ES cell line derived from *Mbd2*$^{-/-}$, *Mbd3*$^{Flox/Flox}$ mice (40, XX) in a mixed C57Black/6 and 129/Ola background. Sall4-FKBP cells were created in *Sall1*$^{-/-}$ *Sall4*$^{+/-}$ ES cells (40, XX) (*Miller et al., 2016*).

Targeting constructs were made using the AID sequence (*Nishimura et al., 2009*) or mini-AID (*Kubota et al., 2013*) amplified from an Oct4-AID plasmid; a gift from José Silva (*Bates et al., 2021*). Targeting plasmids using the FKBP protein (*Nabet et al., 2018*) were made by amplifying FKBP from pLEX_305-C-dTAG; a gift from James Bradner & Behnam Nabet (Addgene plasmid # 91798).

To create Auxin-depletable cell lines, parent lines were first transfected with a PiggyBac construct to constitutively express OsTir1 (created using pMGS56 (GFP-ARF16-PB1-P2A-OsTIR1); a gift from Michael Guertin (Addgene plasmid # 129668)) linked to either G418 or Hygromycin resistance, and cells were cultured under selection to maintain OsTir1 expression. To create degron knock-ins, cells were lipofected with a targeting vector and appropriate gRNA (*Table 1*) in a Cas9-expression vector

**Table 1.** gRNA sequences used for gene targeting.

| Gene | gRNA | Sequence (5'–3') |
| --- | --- | --- |
| Chd4 | gRNA1 | GGTGGAGGTGGATATCACTC |
| Mbd3 A3xF | gRNA1 | TTCTCACGCGTCACTCGCTC |
| Mbd3 | gRNA2 | CAGCCATTCCCTGGAAGTAC |
| Sall4 | gRNA1 | AATAAGATTGCTGTCAGCTA |
| Sall4 | gRNA2 | AAGATTGCTGTCAGCTAAGG |
| Nanog | gRNA1 | AACTACTCTGTGACTCCACC |
| Sox2 | gRNA1 | TGCCCCTGTCGCACATGTGA |
| Klf4 | gRNA2 | GTGGGTCACATCCACTACGT |

(pSpCas9(BB)-2A-GFP (PX458); a gift from Feng Zhang; Addgene plasmid # 48138). Drug-resistant colonies were genotyped and correctly targeted clones were subsequently transiently transfected with an expression plasmid for Dre recombinase to remove ROXed drug selection cassettes. *Chd4* targeting required two rounds of transfection, selection, and drug removal, while *Mbd3* and *Sall4* targeting required one round. For protein depletion, cells were treated with 500 µM Auxin or 500 nM dTAG-13 in standard culture media. All HALOTag-fusion ES cell lines were heterozygous for the HALOTag fusion, that is HALO/+, and verified by western blotting.

## Nuclear extract fractionation and western blotting

Nuclear fractionation was carried out as described (*Gillotin, 2018*). Briefly, cells were collected in ice-cold PBS and pelleted in a refrigerated centrifuge. The cell pellet was lysed by gentle up and down pipetting five times in ice-cold buffer E1 (50 mM HEPES, 140 mM NaCl, 1 mM EDTA, 10% glycerol, 0.5% NP-40, 0.25% Triton X-100, 1 mM DTT, protease inhibitors). After pelleting and washing in E1 buffer, the pellet was resuspended in ice-cold E2 buffer (10 mM Tris-HCl, 200 mM NaCl, 1 mM EDTA, 0.5 mM EGTA, protease inhibitors) and shaken for 45 min at 1400 rpm at 4°C. The supernatant, representing the nuclear fraction, was collected into a fresh tube. After washing in E2 buffer, the pellet was resuspended in ice-cold E3 buffer (50 mM Tris-HCl, 20 mM NaCl, 1 mM $MgCl_2$, 1% NP-40, protease inhibitors). The resuspended pellet was sonicated at 4°C in a Bioruptor Plus (Diagenode) for 5 min, using 30 s ON/30 s OFF cycles at high power. Following sonication, nuclear and chromatin fractions were centrifuged at 16,000 × *g* at 4°C for 10 min. 10 µg of extract per lane was used for western blots. Antibodies are listed in the Key Resources Table.

## Cell cycle analysis

The cell cycle distributions of auxin (500 µM) – treated ES cell lines were assessed using propidium iodide (PI) staining coupled with flow cytometry. Approximately 200,000 single cells were fixed using 70% ethanol for at least 24 hr before staining. Fixed samples were washed twice in PBS and resuspended in 200 µl of 50 µg/ml PI (Invitrogen, P3566) and 50 µg/ml DNase and protease-free RNase A (Thermo Scientific, EN0531), diluted in sterile PBS and incubated overnight at 4°C in the dark. The fluorescent intensity of stained samples was determined using the Attune NxT (Thermo Fisher) equipped with a 561-nm laser line and a minimum of 10,000 single-cell events were recorded. The resulting FCS files were analysed using FlowJo (v10.10) and tested for significance using a mixed-effects model with Dunnett's multiple comparisons correction.

## Single-molecule imaging

ES cells were passaged 24 hr before imaging onto either No 1.0 35 mm glass bottom dishes (MatTek Corporation P35G-1.0-14-C) or No 1.5 35 mm glass bottom dishes (MatTek Corporation P35G-1.5-14-C) with their surfaces pre-coated in poly-L-ornithine (Sigma-Aldrich P4957) for ≥30 min at 37°C, followed by three PBS rinses at room temperature, followed by 100 µg/ml Laminin (Sigma-Aldrich L2020) coating in PBS for >4 hr at room temperature. Cells were labelled on the day of imaging with 250 nM HaloTag-PA-JF646 (a gift from L. Lavis, Janelia) for 15 min, rinsed twice in PBS and incubated for 20 min at 37°C in fresh media. PBS rinsing and 20-min incubation steps were repeated five more times, and the cells were then imaged in fresh media.

Single-molecule tracking was carried out using oblique illumination (*Tokunaga et al., 2008*) on a custom-built double-helix point spread function microscope with a Nikon Eclipse Ti-U inverted microscope body and a box incubator set to 37°C (*Carr et al., 2017*). Beams were expanded and collimated using Galilean beam expanders, then combined using dichroic mirrors. For 2D imaging, a Nikon 1.49 NA 60× oil immersion objective (CFI Apochromat TIRF 60XC Oil) was used to focus excitation beams onto the sample, and the microscope was also set to an internal magnification of ×1.5. For 3D DHPSF imaging, a Nikon 1.27 NA 60x water immersion objective lens (CFI SR Plan Apo IR 60XC WI) was used without internal ×1.5 magnification. The emission path of the microscope was also modified to include a fixed double-helix phase mask (DoubleHelix, Boulder, CO) in the Fourier domain of the emission path of the microscope (*Carr et al., 2017*; *Pavani et al., 2009*). For long-exposure imaging, control samples were fixed in 4% formaldehyde (vol/wt) in PBS for 10 min at room temperature, rinsed twice with PBS and stored in PBS. Samples were imaged in identical pre-warmed culture media and under identical conditions to live-cell samples. Across all experiments, three separate samples per condition

were imaged in a day and all experiments were repeated on at least two separate days for biological replications.

Background subtraction with a five-pixel rolling ball radius was carried out on image stacks using open-source software Fiji (*Schindelin et al., 2012*). Single molecules imaged in 2D were localised using the PeakFit tool within the GDSC single-molecule light microscopy (SMLM) plugin (https://github.com/aherbert/gdsc-smlm; *Herbert, 2026*; *Etheridge et al., 2022*) for Fiji. Localisations were filtered for precision better than 25 nm. For single molecules imaged in 3D, PeakFit was used as above but with an initial precision threshold of 40 nm and an additional analysis step after this. 2D localisations of DHPSF lobes were paired to generate 3D localisations using DHPSFU (https://github.com/TheLaueLab/DHPSFU, copy archived at *Ponjavic and Jartseva, 2025*). Localisations were then tracked across frames to generate trajectories using custom Python code (https://github.com/wb104/trajectory-analysis copy archived at *wb104, 2026*) . Localisations were connected between two successive frames if they were located within 400 nm of each other.

Bound fractions were calculated for a sample based on the number of trajectory frames assigned as being confined, divided by the total number of trajectory frames. Tracks with a residence time shorter than 1.5 s were filtered out to reduce the impact of noise on further analysis. The decay curve of residence times for each sample was fitted to a single exponential fit using MATLAB (2018) to yield apparent dissociation rates. The measured apparent dissociation rate ($k_{off}^{app}$) was calculated by fitting a single exponential decay function to the survival (Kaplan–Meier) function ($S$) of TF dwell times measured from the point of first observation ($t$), as follows:

$$S = e^{-k_{off}^{app}t}.$$

## RNA-seq

Cells were harvested in Trizol Reagent and RNA purified using Zymo Direct-zol columns (Zymo Research) according to the manufacturer's instructions. 60–100 ng of Ribosomal RNA-depleted mRNA was used for library preparation with the NEXTflex RapidDirectional RNA-seq kit (Illumina). 100 bp paired end (2 × 50 bp) sequencing was performed on a NovaSeq S1 flowcell at the CRUK Cambridge Institute sequencing facility.

Raw data were trimmed using TrimGalore v0.6.4. Reads were indexed and aligned to the unmasked GRC38/mm10 reference genome using the Burrows–Wheeler Aligner (BWA) (*Li and Durbin, 2009*). Successful removal of adapter and low-quality bases was assessed using FastQC after trimming. File conversion, sorting, removal of duplicates and mitochondrial reads were performed using samtools (*Li et al., 2009*).

Read counts for genomic features were summarised using featureCounts (*Liao et al., 2014*) and a list of the GRC38/mm10 genomic features in a GTF format downloaded from https://www.ensembl.org/ and https://genome.ucsc.edu/cgi-bin/hgTables. Differential expression was analysed using DESeq2 (*Love et al., 2014*). Raw expression counts were transformed and normalised to FPKM (fragments per kilobase of transcript per million mapped reads). Simple linear models were used for pairwise comparative analyses between the 0-time point and depletion time points. An adjusted p-value threshold of 0.05 was used to identify significantly differentially expressed genes. For principal component analysis and the generation of correlation matrices vst-transformed data were used within DESeq2.

Genes were labelled with their Ensembl and their common gene symbols using the Annotation Hub and ensembldb packages (*Rainer et al., 2019*). The ComplexHeatmap package was used for the generation of expression heatmaps (*Gu et al., 2016*). Volcano plots were generated using the EnhancedVolcano package. To find enriched gene ontology terms among differentially expressed genes, the enrichGO function of the clusterProfiler R package and the genome-wide annotations were retrieved using the biomaRt package (*Yu et al., 2012*).

## ATAC-seq

100,000 nuclei were used for each reaction using the OMNI-ATAC protocol (*Corces et al., 2017*) with 10,000 rat ES cell nuclei added as a spike-in control (a gift from Austin Smith, Exeter) (*Buehr et al., 2008*). Tagmentation was achieved using Tn5 made in-house (*Picelli et al., 2014*) for 30 min at 37°C on a shaking block. Immediately after incubation, DNA was purified using a Zymo DNA Clean and Concentrator Kit (Zymo Research). DNA was then eluted in 21 µl of DNAse/RNAse-free $H_2O$.

Barcoding for library preparation was performed by PCR amplification using NEBNext High-Fidelity 2X PCR Master Mix (New England Biolabs) and NEBNext index primers for Illumina sequencing. Amplified samples were purified using AMPure SPRI beads (VWR International Ltd) and resuspended in 25 µl of 10 mM Tris-HCl pH 8.0. The Agilent 4200 TapeStation System with D1000 ScreenTape and D1000 Reagents (Agilent) was used for quantification and fragmentation check of the samples. Equimolar ratios of all samples (7 conditions × 2 biological replicates × 2 technical replicates each) were pooled for 300 bp paired end (2 × 150 bp) sequencing performed on a NovaSeq S2 flowcell with the CRUK Cancer Institute sequencing facility.

Raw data were trimmed to remove adapter contamination using TrimGalore v0.6.4. Reads were aligned to the GRC38/mm10 reference genome and the *Rattus norvegicus*/Rnor_6.0 genome (spike-in) using the BWA aligner. File conversion, sorting, removal of duplicates and mitochondrial reads were performed using SAMtools (*Li et al., 2009*). For visualisation purposes, bigwig files were generated using bamCoverage from deepTools (*Ramírez et al., 2014*). A scale factor was applied, calculated as the number of uniquely mapped rat spike-in reads for each sample divided by the number of uniquely mapped reads in the sample with the lowest count. All replicates were pooled together into a single track using bigWigMerge from UCSC Tools (*Kent et al., 2002*). Mean profiles of the signal and heatmaps of these regions were plotted using the ComputeMatrix, plotHeatmap and plotProfile functions from deepTools.

Peak calling was performed for ≤120 bp fragments using macs2 (*Zhang et al., 2008*) and applying the -f BAMPE -q 0.05 –nolambda –keep-dup auto parameters. Peak annotation and motif analysis was conducted using the annotatePeaks and findMotifsGenome functions from HOMER (*Heinz et al., 2010*). Integration of these annotated peaks with RNA-seq gene expression data was conducted using custom R scripts. Differential accessibility analysis was carried out using DiffBind (https://bioconductor.org/packages/DiffBind). Differentially accessible regions were identified in pairwise comparisons to time point-0 using the rat aligned reads as a spike-in and an adjusted p-value threshold of 0.05. Chromatin state enrichment analysis for differentially accessible regions was performed using ChromHMM (*Ernst and Kellis, 2017*), using a predefined chromatin state model generated from E14 mouse ESCs ChIP-seq data (*Pintacuda et al., 2017*).

To analyse the Tn5 integration sites of the ATAC-seq data, the plotFootprint() function in VplotR (*Serizay and Ahringer, 2021*) was used with the addition of a code in a loop to calculate the normalised version of the cuts by manually dividing by the library size of each merged bam file. To analyse the fragment sizes of the ATAC-seq data that correspond to different nucleosome structure sizes, the plotVmat() function in VplotR was used. The merged reads of different time points of CHD4 depletion were normalised using the native libdepth + nloci option, which normalises the reads using the library depth and the number of loci.

## Cut&Run and Cut&Tag

Cut&Run and Cut&Tag were carried out as described (*Janssens et al., 2022*; *Meers et al., 2019*). For Cut&Run, 100,000 live ES cells were used per reaction, with 10,000 rat nuclei added as a spike-in control. Cut&Tag was performed on 100,000 nuclei per reaction. Antibodies (Key Resources Table) were used at 1/100 dilution. pAG-MNAse and pA-Tn5 were made and purified in-house as described (*Kaya-Okur et al., 2019*; *Meers et al., 2019*). Plasmids 3XFlag-pA-Tn5-Fl and pAG/MNase were a gift from Steven Henikoff (Addgene plasmids #124601 and #123461, respectively). Library preparation for sequencing was carried out in the CSCI Genomics facility. 300 bp paired-end sequencing was performed on a NovaSeqX 25B flow cell at the CRUK Cancer Institute sequencing facility.

Raw data were trimmed to remove adapter contamination using TrimGalore v0.6.4. Reads were aligned to the unmasked GRC38/mm10 reference genome and the *Rattus norvegicus*/Rnor_6.0 genome (spike-in) using the BWA. File conversion, sorting, removal of duplicates, and mitochondrial reads were performed using SAMtools. For visualisation purposes, bigwig files were generated using bamCoverage from deepTools. A scale factor was applied, calculated as the number of uniquely mapped rat spike-in reads for each sample divided by the number of uniquely mapped reads in the sample with the lowest count. All replicates were pooled together into a single track using bigWigMerge from UCSC Tools. Mean profiles of the signal and heatmaps of these regions were plotted from these tracks using the ComputeMatrix, plotHeatmap and plotProfile functions from deepTools. Peak calling was performed using macs2 and SEACR (*Meers et al., 2019*). SEACR was used in non-control

mode with the stringent threshold setting (threshold = 0.01). Input BED files consisted of scaled fragment bedgraphs generated from Cut&Run data. Scaling was performed using a factor based on the number of uniquely mapped rat spike-in reads for each sample, which was then adjusted relative to the sample with the fewest mapped reads. Motif analysis was conducted using HOMER. Differential accessibility analysis was also carried out using the DiffBind package, incorporating DESeq2 functionality. Differentially bound peaks were identified in pairwise comparisons to time point-0 using an adjusted p-value threshold of 0.05. Chromatin state enrichment analysis for differentially bound regions was performed using ChromHMM as for ATAC-seq.

## MNase-seq

All MNase experiments, sequencing and data processing were carried out exactly as described (*Bornelöv et al., 2018*) except that nuclei were digested with 500 U/ml MNase (New England Biolabs) at 24°C for 15 min with shaking.

## Accession numbers

ATAC-seq: E-MTAB-15037, E-MTAB-15375
RNA-seq: E-MTAB-15102
nascent RNA-seq: E-MTAB-15127
Cut&Run: E-MTAB-15606, E-MTAB-15607, GSE311420, GSE203303 (*Ru et al., 2022*)
Cut&Tag: E-MTAB-15625, E-MTAB-15627
MNase: E-MTAB-6807 (*Bornelöv et al., 2018*), PRJNA1332303

## Materials availability statement

Plasmids created as part of this study are available from Addgene: https://www.addgene.org/Brian_Hendrich/. Cell lines can be requested from BDH.

## Acknowledgements

We are grateful to Nick Owens, Vladimir Teif, Joel Mackay, and past and present Hendrich and Laue group members for helpful discussions; to David Klenerman and Ziwei Zhang for help with imaging; to Marco Trizzino for support and to the CSCI facilities for expert technical assistance. This work was supported by PhD studentships from Wolfson College, Cambridge to AK, from AstraZeneca to OO and from the Medical Research Council to DS, and by grants from the Medical Research Council to BDH (MR/X018342/1 and MR/Y000595/1) and to EDL (MR/P019471/1 and MR/M010082/1), from the Wellcome Trust to EDL (206291/Z/17/Z), from the Isaac Newton Trust to BDH (17.24(aa)), and core funding from the Wellcome/MRC (203151/Z/16/Z) to the Cambridge Stem Cell Institute.

---

## Additional information

### Funding

| Funder | Grant reference number | Author |
|---|---|---|
| Medical Research Council | MR/Y000595/1 | Maya Lopez Nicola Reynolds Brian Hendrich |
| Medical Research Council | MR/X018342/1 | India Baker Brian Hendrich |
| Wolfson College, University of Cambridge | | Andria Koulle |
| AstraZeneca | | Oluwaseun Ogundele |
| Medical Research Council | | Devina Shah |
| Medical Research Council | MR/P019471/1 | Ernest D Laue |

| Funder | Grant reference number | Author |
|---|---|---|
| Medical Research Council | MR/M010082/1 | Ernest D Laue |
| Wellcome | 206291/Z/17/Z | Nicola Reynolds<br>Ernest D Laue<br>Brian Hendrich |
| Isaac Newton Trust | 17.24(aa) | Brian Hendrich |
| Wellcome - MRC Cambridge Stem Cell Institute, University of Cambridge | 203151/Z/16/Z | Andria Koulle<br>Oluwaseun Ogundele<br>Maya Lopez<br>Nicola Reynolds<br>Brian Hendrich |

The funders had no role in study design, data collection, and interpretation, or the decision to submit the work for publication. For the purpose of Open Access, the authors have applied a CC BY public copyright license to any Author Accepted Manuscript version arising from this submission.

## Author contributions

Andria Koulle, Data curation, Investigation, Methodology, Formal analysis, Writing – review and editing; Oluwaseun Ogundele, Data curation, Formal analysis, Methodology; Devina Shah, Resources, Data curation, Formal analysis, Validation, Investigation, Methodology, Writing – review and editing; India Baker, Data curation, Formal analysis, Investigation, Methodology; Maya Lopez, Data curation, Software, Formal analysis; David Lando, Investigation; Nicola Reynolds, Data curation, Formal analysis, Supervision, Investigation, Methodology, Writing – review and editing; Ramy Ragheb, Data curation, Software, Formal analysis, Supervision, Methodology, Writing – review and editing; Ernest D Laue, Conceptualization, Data curation, Formal analysis, Supervision, Funding acquisition, Validation, Writing – review and editing; Brian Hendrich, Conceptualization, Data curation, Formal analysis, Funding acquisition, Methodology, Writing – original draft, Project administration, Writing – review and editing

## Author ORCIDs

Andria Koulle (iD) https://orcid.org/0009-0004-0321-2858
Oluwaseun Ogundele (iD) https://orcid.org/0000-0003-1979-6101
Devina Shah (iD) https://orcid.org/0009-0004-1944-993X
India Baker (iD) https://orcid.org/0000-0002-4787-6259
Maya Lopez (iD) https://orcid.org/0009-0006-4806-1103
David Lando (iD) https://orcid.org/0000-0001-5783-8769
Nicola Reynolds (iD) https://orcid.org/0000-0003-1620-2750
Ramy Ragheb (iD) https://orcid.org/0000-0001-5892-1767
Ernest D Laue (iD) https://orcid.org/0000-0002-7476-4148
Brian Hendrich (iD) https://orcid.org/0000-0002-0231-3073

Reviewer #1 (Public review): https://doi.org/10.7554/eLife.109280.3.sa1
Reviewer #2 (Public review): https://doi.org/10.7554/eLife.109280.3.sa2
Reviewer #3 (Public review): https://doi.org/10.7554/eLife.109280.3.sa3
Author response https://doi.org/10.7554/eLife.109280.3.sa4

# Additional files

## Supplementary files
MDAR checklist

## Data availability
High throughput sequencing data have been deposited in Array Express: ATAC-seq: E-MTAB-15037 RNAseq: E-MTAB-15102 nascent RNAseq: E-MTAB-15127 Cut&Run: E-MTAB-15606, E-MTAB-15607 Cut&Tag: E-MTAB-15625 Sall4 Depletion ATAC-seq: E-MTAB-15375 Sall4, in GEO: Cut&Run: GSE311420 and in SRA: MNase: PRJNA1332303.

The following datasets were generated:

| Author(s) | Year | Dataset title | Dataset URL | Database and Identifier |
|---|---|---|---|---|
| Hendrich B | 2025 | ATAC-seq data of mouse embryonic stem cells treated with auxin for CHD4 depletion against untreated control | https://www.ebi.ac.uk/biostudies/ArrayExpress/studies/E-MTAB-15037?query=E-MTAB-15037 | ArrayExpress, E-MTAB-15037 |
| Hendrich B, Koulle A | 2025 | RNA-seq data of mouse embryonic stem cells treated with auxin for CHD4 depletion against untreated control | https://www.ebi.ac.uk/biostudies/ArrayExpress/studies/E-MTAB-15102 | ArrayExpress, E-MTAB-15102 |
| Hendrich B, Koulle A | 2025 | Nascent RNA-seq data of mouse embryonic stem cells treated with auxin for CHD4 depletion against untreated control | https://www.ebi.ac.uk/biostudies/ArrayExpress/studies/E-MTAB-15127?query=E-MTAB-15127 | ArrayExpress, E-MTAB-15127 |
| Hendrich B, Koulle A | 2026 | CUT&RUN data of Nanog and Sox2 in mouse embryonic stem cells treated with auxin for CHD4 depletion against untreated control | https://www.ebi.ac.uk/biostudies/ArrayExpress/studies/E-MTAB-15607 | ArrayExpress, E-MTAB-15607 |
| Hendrich B, Koulle A | 2025 | CUT&Tag data of H3K27Ac and H3K4Me1 in mouse embryonic stem cells treated with auxin for CHD4 depletion against untreated control | https://www.ebi.ac.uk/biostudies/ArrayExpress/studies/E-MTAB-15625 | ArrayExpress, E-MTAB-15625 |
| Hendrich B, Koulle A | 2025 | ATAC-seq data of mouse embryonic stem cells treated with dTAG for SALL4 depletion against untreated control | https://www.ebi.ac.uk/biostudies/ArrayExpress/studies/E-MTAB-15375 | ArrayExpress, E-MTAB-15375 |
| Hendrich B, Koulle A | 2026 | CUT&RUN data of Chd4 and Mbd3 in mouse embryonic stem cells | https://www.ebi.ac.uk/biostudies/ArrayExpress/studies/E-MTAB-15606 | ArrayExpress, E-MTAB-15606 |
| Ma X, Lando D, Laue E | 2026 | Single cell genome structures of mouse diploid ES cells [CUT&Run] | https://www.ncbi.nlm.nih.gov/geo/query/acc.cgi?acc=GSE311420 | NCBI Gene Expression Omnibus, GSE311420 |
| Ragheb R, Reynolds N, Hendrich B | 2026 | The chromatin remodeller CHD4 controls both nucleosome integrity and transcription factor binding to promote activity of active regulatory elements and to prevent activation of silent enhancers | https://www.ncbi.nlm.nih.gov/bioproject/PRJNA1332303/ | NCBI BioProject, PRJNA1332303 |

The following previously published datasets were used:

| Author(s) | Year | Dataset title | Dataset URL | Database and Identifier |
|---|---|---|---|---|
| Ru et al., Ru W, Koga T, Wang X, Guo Q | 2022 | SALL4 Occupany in mESCs by Cut & Run | https://www.ncbi.nlm.nih.gov/geo/query/acc.cgi?acc=GSE203303 | NCBI Gene Expression Omnibus, GSE203303 |
| Berton P | 2017 | Nucleosome occupancy during NuRD complex induction | https://www.ebi.ac.uk/biostudies/ArrayExpress/studies/E-MTAB-6807 | ArrayExpress, E-MTAB-6807 |

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

# Appendix 1

## Appendix 1—key resources table

| Reagent type (species) or resource | Designation | Source or reference | Identifiers | Additional information |
|---|---|---|---|---|
| Cell line (*Mus musculus*) (XY) | BC8 ES cells | PMID:26025256 | | F1 hybrid from a C57Black/6 and Mus castaneus cross (40, XY), obtained from Anne Ferguson-Smith |
| Cell line (*Mus musculus*) (XY) | CHD4-mAID ADNP-FKBP ES cells | This paper | | CHD4-mAID/ADNP-FKBP/OsTir1 double depletable cell line. G418 resistant. Can be requested from BDH |
| Cell line (*Mus musculus*) (XX) | MBD3-AID ES cells | This paper | | MBD3-AID/OsTir1. G418, Hyg resistant. Made in 23AF. Can be requested from BDH |
| Cell line (*Mus musculus*)(XX) | 23AF ES cells | This paper | | primary ES cell line derived from *Mbd2*$^{-/-}$, *Mbd3*$^{Flox/Flox}$ mice. G418 resistant. Can be requested from BDH |
| Cell line (*Mus musculus*) (XX) | Sall4-FKBP ES cells | This paper | | Created in Sall1$^{-/-}$ Sall4$^{+/-}$ ES cells. Can be requested from BDH |
| Cell line (*Mus musculus*) (XX) | Sall1$^{-/-}$ Sall4$^{+/-}$ ES cells | PMID:27471257 | | Parent line for Sall4-FKBP cells |
| Cell line (*Mus musculus*)(XX) | Mbd3-inducible ES cells | PMID:30008319 | | MER-MBD3b-MER in Mbd3(−/−) ES cells |
| Cell line (*Rattus norvegicus*) | Rat ES cells | PMID:19109897 | | Used as spike-in for genomics experiments. A gift from Austin Smith |
| Antibody | anti-CHD4, mouse monoclonal | Abcam | RRID:AB_2229454, ab70469 | (1/5000) |
| Antibody | anti-CHD4, rabbit polyclonal | Abcam | RRID:AB_1268107, ab72418 | (1/10,000) |
| Antibody | anti-FLAG, mouse monoclonal | Sigma-Aldrich | RRID:AB_259529/RRID:AB_262044, F3165/F1804 | (1/2000) |
| Antibody | anti-HA, mouse monoclonal | Invitrogen | RRID:AB_10978021, 26183 | (1/2000) |
| Antibody | anti-ADNP, goat polyclonal | R&D Systems | AF5919 | (1/2000) |
| Antibody | anti-Histone H3, rabbit polyclonal | Abcam | RRID:AB_302613, ab1791 | (1/5000) |
| Antibody | anti-MTA2, mouse monoclonal | Abcam | RRID:AB_2146939, ab50209 | (1/5000) |
| Antibody | anti-NANOG, rabbit polyclonal | Bethyl Labs | RRID:AB_386108, A300-387A | (1/2000) |
| Antibody | anti-SOX2, rat monoclonal | Ebioscience | RRID:AB_11219471, 14-9811-82 | (1/2000) |
| Antibody | anti-LaminB1, rabbit polyclonal | Abcam | RRID:AB_2616597, ab133741 | (1/10,000) |
| Antibody | anti-H3K27Ac, rabbit polyclonal | Abcam | RRID:AB_2118291, ab4729 | (1/1000) |
| Antibody | anti-H3K4Me1, rabbit polyclonal | Abcam | RRID:AB_306847, ab8895 | (1/1000) |
| Antibody | anti-H3K4Me3, rabbit polyclonal | Millipore | RRID:AB_1163444, 04-745 | (1/1000) |
| Antibody | anti-H3K27Me3, rabbit polyclonal | Millipore | RRID:AB_310624, 07-449 | (1/1000) |
| Recombinant DNA reagent | Oct4-AID | PMID:34143975 | | AID source. A gift from José Silva |

*Appendix 1 Continued on next page*

*Appendix 1 Continued*

| Reagent type (species) or resource | Designation | Source or reference | Identifiers | Additional information |
|---|---|---|---|---|
| Recombinant DNA reagent | pLEX_305-C-dTAG | Addgene | RRID:Addgene_91798 | FKBP source.https://www.addgene.org/ |
| Recombinant DNA reagent | pMGS56 | Addgene | RRID:Addgene_129668 | GFP-ARF16-PB1-P2A-OsTIR1. https://www.addgene.org/ |
| Recombinant DNA reagent | pSpCas9(BB)-2A-GFP | Addgene | RRID:Addgene_48138 | Cas9/gRNA expression.https://www.addgene.org/ |
| Recombinant DNA reagent | PB CAG Tir1 iHyg | This paper | RRID:Addgene_235506 | OsTir1 Expression construct. https://www.addgene.org/Brian_Hendrich/ |
| Recombinant DNA reagent | pBS_GGSG_Fkbp_2xHA_BB2 | This paper | RRID:Addgene_235503 | FKBP Cloning Vector. https://www.addgene.org/Brian_Hendrich/ |
| Recombinant DNA reagent | pBS_GGSG_mAID_HA_BactBsd | This paper | RRID:Addgene_235504 | mAID Cloning Vector. https://www.addgene.org/Brian_Hendrich/ |
| Recombinant DNA reagent | pBS AID-3xF_PP | This paper | RRID:Addgene_235502 | AID Cloning Vector. https://www.addgene.org/Brian_Hendrich/ |
| Recombinant DNA reagent | pMbd3-GGSG-AID-3xF_PP | This paper | RRID:Addgene_235507 | MBD3-AID targeting vector. https://www.addgene.org/Brian_Hendrich/ |
| Recombinant DNA reagent | pChd4 GGSG_mAID-3xF_BactBsd | This paper | RRID:Addgene_235508 | CHD4-mAID targeting vector. https://www.addgene.org/Brian_Hendrich/ |
| Recombinant DNA reagent | pSall4_GGSG_Fkbp_2xHA_BB | This paper | RRID:Addgene_235510 | Sall4-FKBP targeting vector. https://www.addgene.org/Brian_Hendrich/ |
| Recombinant DNA reagent | pAdnp_GGSG_Fkbp_2xHA_BB | This paper | RRID:Addgene_235526 | Sall4-FKBP targeting vector. https://www.addgene.org/Brian_Hendrich/ |
| Recombinant DNA reagent | pNanog-Halo-Ty2-BactBsd | This paper | RRID:Addgene_235511 | Nanog-Halo targeting vector. https://www.addgene.org/Brian_Hendrich/ |
| Recombinant DNA reagent | pKlf4_Halo_Ty2_BactBsd | This paper | RRID:Addgene_235512 | Klf4-Halo targeting vector. https://www.addgene.org/Brian_Hendrich/ |
| Recombinant DNA reagent | pSox2_Halo_Ty2_BactBsd | This paper | RRID:Addgene_235513 | Sox2-Halo targeting vector. https://www.addgene.org/Brian_Hendrich/ |
| Recombinant DNA reagent | 3XFlag-pA-Tn5-Fl | Addgene | RRID:Addgene_124601 | https://www.addgene.org/ |
| Recombinant DNA reagent | pAG/MNase | Addgene | RRID:Addgene_123461 | https://www.addgene.org/ |
| Sequence-based reagent | ATAC-seq: 0, 30, 60, and 240 min CHD4 depletion | This paper | E-MTAB-15037 | https://www.ebi.ac.uk/biostudies/arrayexpress/studies/ |
| Sequence-based reagent | ATAC-seq: 0, 1, 4, and 8 hr SALL4 depletion | This paper | E-MTAB-15375 | https://www.ebi.ac.uk/biostudies/arrayexpress/studies/ |
| Sequence-based reagent | RNA-seq: 0, 1, 2, 4, and 24 hr CHD4 depletion | This paper | E-MTAB-15102 | https://www.ebi.ac.uk/biostudies/arrayexpress/studies/ |
| Sequence-based reagent | nascent RNA-seq: 0, 1, 2, 3, 4, and 6 hr CHD4 depletion | This paper | E-MTAB-15127 | https://www.ebi.ac.uk/biostudies/arrayexpress/studies/ |

*Appendix 1 Continued on next page*

*Appendix 1 Continued*

| Reagent type (species) or resource | Designation | Source or reference | Identifiers | Additional information |
|---|---|---|---|---|
| Sequence-based reagent | Cut&Run: MBD3 and CHD4 | This paper | E-MTAB-15606 | https://www.ebi.ac.uk/biostudies/arrayexpress/studies/ |
| Sequence-based reagent | Cut&Run: NANOG and SOX2 after 0, 30, 60, and 240 min of CHD4 depletion | This paper | E-MTAB-15607 | https://www.ebi.ac.uk/biostudies/arrayexpress/studies/ |
| Sequence-based reagent | Cut&Run: SALL4 | PMID:36257403 | GSE203303 | https://www.ncbi.nlm.nih.gov/gds |
| Sequence-based reagent | Cut&Run: H3K4Me3 and H3K27Me3 | This paper | GSE311420 | https://www.ncbi.nlm.nih.gov/gds |
| Sequence-based reagent | Cut&Tag: H3K27Ac and H3K4Me1 after 0, 4, and 24 hr of CHD4 depletion | This paper | E-MTAB-15625 | https://www.ebi.ac.uk/biostudies/arrayexpress/studies/ |
| Sequence-based reagent | Cut&Tag: MBD3 after 0, 4, and 24 hr CHD4 depletion | This paper | E-MTAB-15627 | https://www.ebi.ac.uk/biostudies/arrayexpress/studies/ |
| Sequence-based reagent | MNase-seq, Mbd3-inducible ES cells, time 0, 0.5, and 24 hr, hard digest | PMID:30008319 | E-MTAB-6807 | https://www.ebi.ac.uk/biostudies/arrayexpress/studies/ |
| Sequence-based reagent | MNase-seq, Mbd3-inducible ES cells, time 0, 0.5, and 24 hr, soft digest | This paper | PRJNA1332303 | https://www.ncbi.nlm.nih.gov/sra |
| Peptide, recombinant protein | Micrococcal nuclease | New England Biolabs | M0247S | |
| Peptide, recombinant protein | pAG-MNAse | PMID:31232687 | | Made in-house |
| Peptide, recombinant protein | pA-Tn5 | PMID:31036827 | | Made in-house |
| Peptide, recombinant protein | Tn5 | PMID:25079858 | | Made in-house |
| Peptide, recombinant protein | mouse LIF | Cambridge Department of Biochemistry | | (10 ng/ml) |
| Commercial assay or kit | Direct-zol RNA Microprep | Zymo Research | R2062 | |
| Commercial assay or kit | DNA Clean and Concentrator-5 | Zymo Research | D4003 | |
| Commercial assay or kit | NEBNext High-Fidelity 2X PCR Master Mix | New England Biolabs | M0541L | |
| Chemical compound, drug | dTAG-13 | Bio-Techne Ltd | 6605/5 | (500 nM) |
| Chemical compound, drug | Auxin | Cambridge Bioscience | 16954-1g-CAY | (500 µM) |
| Chemical compound, drug | HaloTag-PA-JF646 | Lavis, Janelia | | (250 nM) |
| Chemical compound, drug | Propidium iodide | Invitrogen | P3566 | |
| Chemical compound, drug | PD0325901 | Cambridge Department of Biochemistry | | (1 mM) |

*Appendix 1 Continued on next page*

*Appendix 1 Continued*

| Reagent type (species) or resource | Designation | Source or reference | Identifiers | Additional information |
|---|---|---|---|---|
| Chemical compound, drug | CHIR99021 | Cambridge Department of Biochemistry | | (3 mM) |
| Software, algorithm | PeakFit tool within the GDSC single-molecule light microscopy (SMLM) plugin, V1 | PMID:37351368 | RRID:SCR_022717 | https://github.com/aherbert/gdsc-smlm |
| Software, algorithm | DHPSFU | PMID:40835649 | | https://github.com/TheLaueLab/DHPSFU |
| Software, algorithm | Trajectory Analysis | PMID:29955052 | | hhttps://github.com/wb104/trajectory-analysis |
| Software, algorithm | R version 4.3.1 (2023-06-16) | https://www.R-project.org/ | RRID:SCR_001905 | https://www.R-project.org/ |
| Software, algorithm | DiffBind | PMID:25972895 | RRID:SCR_012918 | https://bioconductor.org/packages/release/bioc/html/DiffBind.html |
| Software, algorithm | DEseq2 | PMID:25516281 | RRID:SCR_015687 | https://bioconductor.org/packages/release/bioc/html/DESeq2.html |
| Software, algorithm | deepTools | PMID:27079975 | RRID:SCR_016366 | https://deeptools.readthedocs.io/ |
| Software, algorithm | VplotR | doi: 10.18129/B9.bioc.VplotR | | https://bioconductor.org/packages/VplotR |
| Software, algorithm | IGV – Integrative genomics viewer | PMID:21221095 | RRID:SCR_011793 | https://igv.org/ |
| Software, algorithm | TrimGalore v0.6.4 | doi: 10.5281/zenodo.5127898 | RRID:SCR_011847 | https://zenodo.org/records/7598955 |
| Software, algorithm | Burrows–Wheeler Aligner (BWA) | PMID:19451168 | RRID:SCR_010910 | https://bio-bwa.sourceforge.net/ |
| Software, algorithm | samtools | PMID:33590861 | RRID:SCR_002105 | https://www.htslib.org/ |
| Software, algorithm | featureCounts | PMID:24227677 | RRID:SCR_012919 | https://subread.sourceforge.net/featureCounts.html |
| Software, algorithm | MACS2 V2 | PMID:18798982 | RRID:SCR_013291 | https://github.com/macs3-project |
| Software, algorithm | HOMER | PMID:20513432 | RRID:SCR_010881 | http://homer.ucsd.edu/homer/ |
| Software, algorithm | ChromHMM | PMID:29120462 | RRID:SCR_018141 | https://ernstlab.biolchem.ucla.edu/software-and-resources/chromhmm |
| Software, algorithm | SEACR v1.3 | PMID:31300027 | RRID:SCR_027011 | https://github.com/FredHutch/SEACR |

