## [Editor Report · eLife Assessment]

This work offers **important** insights into the protein CHD4's function in chromatin remodeling and gene regulation in embryonic stem cells, supported by extensive biochemical, genomic, and imaging data. The use of an inducible degron system allows precise functional analysis, and the datasets generated represent a key resource for the field. The revised study offers **compelling** evidence and makes a significant contribution to understanding CHD4's role in epigenetic regulation. This work will be of interest to the epigenetics and stem biology fields.

---

## [Referee Report · Reviewer #1 (Public review)]

Summary:

The authors performed an elegant investigation to clarify the roles of CHD4 in chromatin accessibility and transcription regulation. In addition to the common mechanisms of action through nucleosome repositioning and opening of transcriptionally active regions, the authors considered here a new angle of CHD4 action through modulating the off rate of transcription factor binding. Their suggested scenario is that the action of CHD4 is context-dependent and is different for highly-active regions vs low-accessibility regions.

Strengths:

This is a very well-written paper that will be of interest to researchers working in this field. The authors performed large work with different types of NGS experiments and the corresponding computational analyses. The combination of biophysical measurements of the off-rate of protein-DNA binding with NGS experiments is particularly commendable.

Comments on revised version:

The authors have addressed all my points

---

## [Referee Report · Reviewer #2 (Public review)]

This study leverages acute protein degradation of CHD4 to define its role in chromatin and gene regulation. Previous studies have relied on KO and/or RNA interference of this essential protein and as such are hampered by adaptation, cell population heterogeneity, cell proliferation and indirect effects. The authors have established an AID2-based method to rapidly deplete the dMi-2 remodeller to circumvent these problems. CHD4 is gone within an hour, well before any effects on cell cycle or cell viability can manifest. This represents an important technical advance that, for the first time, allows a comprehensive analysis of the immediate and direct effect of CHD4 loss of function on chromatin structure and gene regulation.

Rapid CHD4 degradation is combined with ATAC-seq, CUT&RUN, (nascent) RNA-seq and single molecule microscopy to comprehensively characterise the impact on chromatin accessibility, histone modification, transcription and transcription factor (NANOG, SOX2, KLF4) binding in mouse ES cells.

The data support the previously developed model that high levels of CHD4/NuRD maintain a degree of nucleosome density to limit TF binding at open regulatory regions (e.g. enhancers). The authors propose that CHD4 activity at these sites is an important prerequisite for enhancers to respond to novel signals that require an expanded or new set of TFs to bind.

What I find even more exciting and entirely novel is the finding that CHD4 removes TFs from regions of limited accessibility to repress cryptic enhancers and to suppress spurious transcription. These regions are characterised by low CHD4 binding and have so far never been thoroughly analysed. The authors correctly point out that the general assumption that chromatin regulators act on regions where they seem to be concentrated (i.e. have high ChIP-seq signals) runs the risk of overlooking important functions elsewhere. This insight is highly relevant beyond the CHD4 field and will prompt other chromatin researchers to look into low level binding sites of chromatin regulators.

The biochemical and genomic data presented in this study is of high quality (I cannot judge single microscopy experiments due to my lack of expertise). This is an important and timely study that is of great interest to the chromatin field.

Comments on revised version:

All my comments below have been addressed in the revised version of the manuscript.

The revised manuscript provides a significant advance of our understanding of how the nucleosome remodeler CHD4 exerts its function. In particular, the findings suggest an intriguing role of CHD4 in TF removal at genomic regions where only low levels of CHD4 can be detected. In the future, it will be interesting to see if this activity is shared by other ATP-dependent nucleosome remodelers.

---

## [Referee Report · Reviewer #3 (Public review)]

Summary:

In this manuscript an inducible degron approach is taken to investigate the function of the CHD4 chromatin remodelling complex. The cell lines and approaches used are well thought out and the data appear to be of high quality. They show that loss of CHD4 results in rapid changes to chromatin accessibility at thousands of sites. At the majority of locations where changes are detected, chromatin accessibility is decreased and these sites are strongly bound by CHD4 prior to activation of the degron and so likely represent primary sites of action. Somewhat surprisingly while chromatin accessibility is reduced at these sites transcription factor occupancy is little changed. Following CHD4 degradation occupancy of the key pluripotency transcription factors NANOG and SOX2 increases at many locations genome wide and at many of these sites chromatin accessibility increases. These represent important new insights into the function of CHD4 complexes.

Strengths:

The experimental approach is well suited to providing insight into a complex regulator such as CHD4. The data generated to characterise how cells respond to loss of CHD4 is of high quality. The study reveals major changes in transcription factor occupancy following CHD4 depletion.

Weaknesses:

The main weakness can be summarised as relating to the fact authors favour the interpretation that all rapid changes following CHD4 degradation occur as a direct effect of the loss of CHD4 activity. The possibility that rapid indirect effects arise does not appear to have been given sufficient consideration. This is especially pertinent where effects are reported at sites where CHD4 occupancy is initially very low (e.g sites where accessibility is gained, in comparison to that at sites where chromatin acdessibility is lost). The revised discussion acknowledges rapid indirect effects cannot be excluded.

---

## [Author Response]

The following is the authors’ response to the original reviews.

**Public Reviews:**

**Reviewer #1 (Public review)**
(1) It might be good to further discuss potential molecular mechanisms for increasing the TF off rate (what happens at the mechanistic level).

This is now expanded in the Discussion

(2) To improve readability, it would be good to make consistent font sizes on all figures to make sure that the smallest font sizes are readable.

We have normalised figure text as much as is feasible.

(3) upDARs and downDARs - these abbreviations are defined in the figure legend but not in the main text.

We have removed references to these terms from the text and included a definition in the figure legend.

(4) Figure 3B - the on-figure legend is a bit unclear; the text legend does not mention the meaning of "DEG".

We have removed this panel as it was confusing and did not demonstrate any robust conclusion.

(5) The values of apparent dissociation rates shown in Figure 5 are a bit different from values previously reported in literature (e.g., see Okamoto et al., 20203, PMC10505915). Perhaps the authors could comment on this. Also, it would be helpful to add the actual equation that was used for the curve fitting to determine these values to the Methods section.

We have included an explanation of the curve fitting equation in the Methods as suggested.

The apparent dissociation rate observed is a sum of multiple rates of decay – true dissociation rate (*k_off_*), signal loss caused by photobleaching *k_pb_*, and signal loss caused by defocusing/tracking error (*k_tl_*).

*k_off_^app^ = k_off_+ k_pb_ + k_tl_*

We are making conclusions about relative changes in *k_off_^app^* upon CHD4 depletion, not about the absolute magnitude of true in *k_off_* or TF residence times.Our conclusions extend to true in *k_off_* on the assumption that *k_pb_* and *k_tl_* are equal across all samples imaged due to identical experimental conditions and analysis. *k_pb_* and *k_tl_* vary hugely across experimental set-ups, especially with different laser powers, so other *k_off_* or *k_off_^app^* values reported in the literature would be expected to differ from ours. Time-lapse experiments or independent determination of *k_pb_* (and *k_tl_*) would be required to make any statements about absolute values of *k_off_*

(6) Regarding the discussion about the functionality of low-affinity sites/low accessibility regions, the authors may wish to mention the recent debates on this (https://www.nature.com/articles/s41586-025-08916-0; https://www.biorxiv.org/content/10.1101/2025.10.12.681120v1).

We have now included a discussion of this point and referenced both papers.

(7) It may be worth expanding figure legends a bit, because the definitions of some of the terms mentioned on the figures are not very easy to find in the text.

We have endeavoured to define all relevant terms in the figure legends.

**Reviewer #2 (Public review):**
(1) Figure 2 shows heat maps of RNA-seq results following a time course of CHD4 depletion (0, 1, 2 hours...). Usually, the red/blue colour scale is used to visualise differential expression (fold-difference). Here, genes are coloured in red or blue even at the 0-hour time point. This confused me initially until I discovered that instead of folddifference, a z-score is plotted. I do not quite understand what it means when a gene that is coloured blue at the 0-hour time point changes to red at a later time point. Does this always represent an upregulation? I think this figure requires a better explanation.

The heatmap displays z-scores, meaning expression for each gene has been centred and scaled across the entire time course. As a result, time zero is not a true baseline, it simply shows whether the gene’s expression at that moment is above or below its own mean. A transition from blue to red therefore indicates that the gene increases relative to its overall average, which typically corresponds to upregulation, but it doesn’t directly represent fold-change from the 0-hour time point. We have now included a brief explanation of this in the figure legend to make this point clear.

(2) Figure 5D: NANOG, SOX2 binding at the KLF4 locus. The authors state that the enhancers 68, 57, and 55 show a gain in NANOG and SOX2 enrichment "from 30 minutes of CHD4 depletion". This is not obvious to me from looking at the figure. I can see an increase in signal from "WT" (I am assuming this corresponds to the 0 hours time point) to "30m", but then the signals seem to go down again towards the 4h time point. Can this be quantified? Can the authors discuss why TF binding seems to increase only temporarily (if this is the case)?

We have edited the text to more accurately reflect what is going on in the screen shot. We have also replaced “WT” with “0” as this more accurately reflects the status of these cells.

(3) There is no real discussion of HOW CHD4/NuRD counteracts TF binding (i.e. by what molecular mechanism). I understand that the data does not really inform us on this. Still, I believe it would be worthwhile for the authors to discuss some ideas, e.g., local nucleosome sliding vs. a direct (ATP-dependent?) action on the TF itself.

We now include more speculation on this point in the Discussion.

**Reviewer #3 (Public review):**
The main weakness can be summarised as relating to the fact that authors interpret all rapid changes following CHD4 degradation as being a direct effect of the loss of CHD4 activity. The possibility that rapid indirect effects arise does not appear to have been given sufficient consideration. This is especially pertinent where effects are reported at sites where CHD4 occupancy is initially low.

We acknowledge that we cannot definitively say any effect is a direct consequence of CHD4 depletion and have mitigated statements in the Results and Discussion.

**Reviewing Editor Comments:**
I am pleased to say all three experts had very complementary and complimentary comments on your paper - congratulations. Reviewer 3 does suggest toning down a few interpretations, which I suggest would help focus the manuscript on its greater strengths. I encourage a quick revision to this point, which will not go back to reviewers, before you request a version of record. I would also like to take this opportunity to thank all three reviewers for excellent feedback on this paper.

As advised we have mitigated the points raised by the reviewers.

**Reviewer #2 (Recommendations for the authors):**
p9, top: The sentence starting with "Genes increasing in expression after four hours...." is very difficult to understand and should be rephrased or broken up.

We agree. This has been completely re-written.

**Reviewer #3 (Recommendations for the authors):**
Sites of increased chromatin accessibility emerge more slowly than sites of lost chromatin accessibility. Figure 1D, a little increase in accessibility at 30min, but a more noticeable decrease at 30min. The sites of increased accessibility also have lower absolute accessibility than observed at locations where accessibility is lost. This raises the possibility that the sites of increased accessibility represent rapid but indirect changes occurring following loss of CHD4. Consistent with this, enrichment for CHD4 and MDB3 by CUT and TAG is far higher at sites of decreased accessibility. The low level of CHD4 occupancy observed at sites where accessibility increases may not be relevant to the reason these sites are affected. Such small enrichments can be observed when aligning to other genomic features. The authors interpret their findings as indicating that low occupancy of CHD4 exerts a long-lasting repressive effect at these locations. This is one possible explanation; however, an alternative is that these effects are indirect. Perhaps driven by the very large increase in TF binding that is observed following CHD4 degradation and which appears to occur at many locations regardless of whether CHD4 is present.

The reviewer is right to point out that we don’t know what is direct and what is indirect. All we know is that changes happen very rapidly upon CHD4 depletion. The changes in standard ATAC-seq signal appear greater at the sites showing decreased accessibility than those increasing, however the starting points are very different: a small increase from very low accessibility will likely be a higher fold change than a more visible decrease from very high accessibility (Fig. 1D). In contrast, Figure 6 shows a more visible increase in Tn5 integrations at sites increasing in accessibility at 30 minutes than the change in sites decreasing in accessibility at 30 minutes. We therefore disagree that the sites increasing in accessibility are more likely to be indirect targets. In further support of this, there is a rapid increase in MNase resistance at these sites upon MBD3 reintroduction (Fig. 6I), possibly indicating a direct impact of NuRD on these sites.

Substantial changes in Nanog and SOX2 binding are observed across the time course. These changes are very large, with 43k or 78k additional sites detected. How is this possible? Does the amount of these TF's present in cells change? The argument that transient occupancy of CHD4 acts to prevent TF's binding to what is likely to be many 100's of thousands of sites (if the data for Nanog and SOX2 are representative of other transcription factors such as KLF4) seems unlikely.

The large number of different sites identified gaining TF binding is likely to be a reflection of the number of cells being analysed: within the 10^5^-10^6^ cells used for a Cut&Run experiment we detect many sites gaining TF binding. In individual cells we agree it would be unlikely for that many sites to become bound at the same time. We detect no changes in the amounts of Nanog or Sox2 in our cells across 4 hour CHD4 depletion time course. However, we maintain that low frequency interactions of CHD4 with a site can counteract low frequency TF binding and prevent it from stimulating opening of a cryptic enhancer.

While increased TF binding is observed at sites of gained accessibility, the changes in TF occupancy at the lost sites do not progress continuously across the time course. In addition, the changes in occupancy are small in comparison to those observed at the gained sites. The text comments on an increase in SOX2 and Nanog occupancy at 30 min, but there is either no change or a loss by 4 hours. It's difficult to know what to conclude from this.

At sites losing accessibility the enrichment of both Nanog and Sox2 increases at 30 minutes. We suspect this is due to the loss of CHD4’s TF-removal activity. Thereafter the two TFs show different trends: Nanog enrichment then decreases again, probably due to the decrease in accessibility at these sites. Sox2, by contrast, does not change very much, possibly due to its higher pioneering ability. It is true that the amounts of change are very small here, however Cut&Run was performed in triplicate and the summary graphs are plotted with standard error of the mean (which is often too small to see), demonstrating that the detected changes are highly significant. (We neglected to refer to the SEM in our figure legends: this has now been corrected.) At sites where CHD4 maintains chromatin compaction, the amount of transcription factor binding goes from zero or nearly zero to some finite number, hence the fold change is very large. In contrast the changes at sites losing accessibility starts from high enrichment so fold changes are much smaller.

Changes in the diffusive motion of tagged TF's are measured. The data is presented as an average of measurements of individual TF's. What might be anticipated is that subpopulations of TF's would exhibit distinct behaviours. At many locations, occupancy of these TF's are presumably unchanged. At 1 hour, many new sites are occupied, and this would represent a subpopulation with high residence. A small population of TF's would be subject to distinct effects at the sites where accessibility reduces at the onehour time point. The analysis presented fails to distinguish populations of TF's exhibiting altered mobility consistent with the proportion of the TF's showing altered binding.

We agree that there are likely subpopulations of TFs exhibiting distinct binding behaviours, and our modality of imaging captures this, but to distinguish subpopulations within this would require a lot more data.

However, there is no reason to believe that the TF binding at the new sites being occupied at 1 hr would have a difference in residence time to those sites already stably bound by TFs in the wildtype, i.e. that they would exhibit a different limitation to their residence time once bound compared to those sites. We do capture more stably bound trajectories per cell, but that’s not what we’re reporting on - it’s the dissociation rate of those that have already bound in a stable manner at sites where TF occupancy is detected also by ChIP.

The analysis of transcription shown in Figure 2 indicates that high-quality data has been obtained, showing progressive changes to transcription. The linkage of the differentially expressed genes to chromatin changes shown in Figure 3 is difficult to interpret. The curves showing the distance distribution for increased or decreased DARs are quite similar for up- and down-regulated genes. The frequency density for gained sites is slightly higher, but not as much higher as would be expected, given these sites are c6fold more abundant than the sites with lost accessibility. The data presented do not provide a compelling link between the CHD4-induced chromatin changes and changes to transcription; the authors should consider revising to accommodate this. It is possible that much of the transcriptional response even at early time points is indirect. This is not unprecedented. For example, degradation of SOX2, a transcriptional activator, results in both repression and activation of similar numbers of genes https://pmc.ncbi.nlm.nih.gov/articles/PMC10577566/

We agree that these figures do not provide a compelling link between the observed chromatin changes and gene expression changes. That 50K increased sites are, on average, located farther away from misregulated genes than are the 8K decreasing sites highlights that this is rarely going to be a case of direct derepression of a silenced gene, but rather distal sites could act as enhancers to spuriously activate transcription. This would certainly be a rare event, but could explain the low-level transcriptional noise seen in NuRD mutants. We have edited the wording to make this clearer.

The model presented in Figure 7 includes distinct roles at sites that become more or less accessible following inactivation of CHD4. This is perplexing as it implies that the same enzymes perform opposing functions at some of the different sites where they are bound.

Our point is that it does the same thing at both kinds of sites, but the nature of the sites means that the consequences of CHD4 activity will be different. We have tried to make this clear in the text.

At active sites, it is clear that CHD4 is bound prior to activation of the degron and that chromatin accessibility is reduced following depletion. Changes in TF occupancy are complex, perhaps reflecting slow diffusion from less accessible chromatin and a global increase in the abundance of some pluripotency transcription factors such as SOX2 and Nanog that are competent for DNA binding. The link between sites of reduced accessibility and transcription is less clear.At the inactive sites, the increase in accessibility could be driven by transcription factor binding. There is very little CHD4 present at these sites prior to activation of the degron, and TF binding may induce chromatin opening, which could be considered a rapid but indirect effect of the CHD4 degron. The link to transcription is not clear from the data presented, but it would be anticipated that in some cases it would drive activation.

We acknowledge these points and have indicated this possibility in the Results and the Discussion.

No Analysis is performed to identify binding sequences enriched at the locations of decreased accessibility. This could potentially define transcription factors involved in CHD4 recruitment or that cause CHD4 to function differently in different contexts.

HOMER analyses failed to provide any unique insights. The sites going down are highly accessible in ES cells: they have TF binding sites that one would expect in ES cells. The increasing sites show an enrichment for G-rich sequences, which reflects the binding preference of CHD4.